# PROTECTION AGAINST SOURCE INFERENCE ATTACKS IN FEDERATED LEARNING

**Andreas Athanasiou**[*]
TU Delft & Inria
Delft, Netherlands & Palaiseau, France
a.athanasiou@tudelft.nl

**Kangsoo Jung**[*]
Inria
Palaiseau, France
kangsoo.jung84@gmail.com

**Catuscia Palamidessi**
Inria & IPP
Palaiseau, France
catuscia@lix.polytechnique.fr

## ABSTRACT

Federated learning (FL) was initially proposed as a privacy-preserving machine learning paradigm. However, FL has been shown to be susceptible to a series of privacy attacks. Recently, there has been concern about the *source inference attack* (SIA), where an honest-but-curious central server attempts to identify exactly which client owns a given data point which was used in the training phase. Alarmingly, standard gradient obfuscation techniques with differential privacy have been shown to be ineffective against SIAs, at least without severely diminishing the accuracy.

In this work, we propose a defense against SIAs within the widely studied shuffle model of FL, where an honest shuffler acts as an intermediary between the clients and the server. First, we demonstrate that standard naive shuffling alone is insufficient to prevent SIAs. To effectively defend against SIAs, shuffling needs to be applied at a more granular level; we propose a novel combination of parameter-level shuffling with the *residue number system* (RNS). Our approach provides robust protection against SIAs without affecting the accuracy of the joint model and can be seamlessly integrated into other privacy protection mechanisms.

We conduct experiments on a series of models and datasets, confirming that standard shuffling approaches fail to prevent SIAs and that, in contrast, our proposed method reduces the attack's accuracy to the level of random guessing.

## 1 INTRODUCTION

Federated learning (FL) McMahan & Moore (2017) is a machine learning framework that trains a global model across multiple clients without centralizing data. Each client updates the model using its local dataset and sends the updates to a central server, which aggregates them to obtain the global model $W$. The most well-known aggregation function is *FedAvg*, where $W \leftarrow \sum_{i=1}^{n} \frac{N_i}{N} w_i$ for $n$ clients, where each client $i$ has $N_i$ data points and $N = \sum_i N_i$. The initial concept of FL was that it was private by design, as the data of each client was not shared with anyone. Unfortunately, an honest-but-curious central server can launch a series of privacy attacks, as it observes the clients' reported model updates and can use them to infer information. For instance, the server can launch a *membership inference attack* (MIA) Shokri et al. (2017); Carlini et al. (2022); Choquette-Choo et al. (2021) which aims to determine if a particular data point exists in the training dataset of *any* client.

The *source inference attack* (SIA) Hu et al. (2021; 2024) has recently been proposed as a natural extension of the MIA. In an SIA, the adversary (an honest-but-curious central server) tries to determine *exactly which* client owns a given data point. To achieve this, the attacker compares the accuracy of the target data point across the received models before they are aggregated into the global model. This can pose a severe violation of privacy; consider for example a medical model jointly built by

---

[*]Primary authors with equal contribution.

different hospitals to treat some disease and suppose that a hospital A is specialized in cancer cases. If the central server learns from a successful SIA that a patient's data was used by Hospital A, then they can confidently assume that the patient suffers from cancer.

**Defending against SIAs** While MIAs exploit model overfitting, SIAs work by exploiting differences in model predictions across clients due to their heterogeneous data distributions. Model obfuscation techniques such as differential privacy (DP) Dwork et al. (2006); Abadi et al. (2016) have been shown to be inadequate to protect against SIAs, at least without severely deteriorating the accuracy of the joint model Hu et al. (2021; 2024). That is because the level of noise is required to be large to successfully make the different local models indistinguishable from each other.

Another approach is to consider regularization-based defenses, which can be beneficial against MIAs since they reduce overfitting Kaya et al. (2020). However, such defenses are insufficient for SIAs as they do not focus on reducing the distributional differences between clients Hu et al. (2024). To verify, we include an empirical evaluation on all common regularization-based defenses (Appendix F). Our results show that regularization-based defenses have virtually no impact against SIAs.

Defenses against Data Reconstruction Attacks (DRAs) (e.g. Instahide Huang et al. (2020), FedAdOb Gu et al. (2024)) are also ineffective as they focus on preventing gradient inversion. In contrast, SIAs assume that the attacker already knows that a given data point was used in training. We empirically verify that such defenses have no effect in Appendix F. Finally, Hu et al. (2024) investigated whether knowledge distillation techniques (e.g. FedMD) can be a means of defense. FedMD was only able to slightly decrease the SIA success rate; the attack remains well above random guessing.

In this work, we focus on the shuffle model, which assumes the presence of a trusted shuffler. This approach, which provides additional privacy protection by permuting clients' data, has been widely studied in FL. Interestingly, although standard naive shuffling removes the connection between the data owner and their value, this is not enough to protect against SIAs. To show this, we propose a series of attacks aimed at defeating the effect of shuffling by remapping the values back to their original owners (Section 5).

Therefore, a new defense strategy against SIAs is necessary. We aim to satisfy these specifications:

- **S.1: Protection**: The accuracy of SIAs is reduced to the level of random guessing, even in the challenging scenario where clients hold datasets with a high level of dissimilarity.

- **S.2: Integrability**: The solution can be seamlessly integrated as a "module" into existing shuffle-model FL architectures, and is compatible with other privacy mechanisms, such as DP, which protect from other kinds of privacy attacks.

- **S.3: Communication efficiency**: Considering that SIAs are better suited for the cross-silo setting of FL (cf. Section 4) an increase in the communication cost can be tolerated, as long as it remains reasonable.

- **S.4: Maintains model accuracy**: DP does not protect against SIAs unless we pay a high price in accuracy. Hence, typical approaches based on DP-SGD would not be optimal.

- **S.5: Minimal trust assumptions**: Each client does not have to trust any additional entity.

**Contributions** Our contributions are summarized as follows:

- We propose novel *reconstruction attacks* that defeat standard shuffling in FL by remapping the shuffled values back to their original owners. We examine three shuffling methods; model-level, layer-level and parameter-level, and provide a corresponding reconstruction algorithm for each. These attacks enable SIAs within the standard shuffle model of FL, making shuffling alone insufficient (Section 5).

- We propose the first robust defense against SIAs in the shuffle model of FL. We introduce a novel dimension-based granular shuffling method, using the *residue number system* (RNS). The defense reduces attack accuracy to random guessing without affecting joint model accuracy and can be seamlessly integrated into existing shuffle mechanisms (Section 6).

- We conduct experiments on the MNIST and CIFAR-10 with CNN and CIFAR-100 with ResNet-18 which validate our analyses (Section 7).

## 2 RELATED WORK

FL remains vulnerable to various attacks despite the fact that raw data are not disclosed Zhu et al. (2019a); Geiping et al. (2020); Hu et al. (2021; 2024). In Hu et al. (2021; 2024) the authors introduced the concept of SIAs and empirically showed, by performing experiments across a variety of datasets, that model overfitting and data distribution (increased heterogeneity) are the most important factors for making a model susceptible to SIAs.

Shuffling has been widely explored in the literature as a means to protect privacy. A shuffler can be implemented distributively using *trusted hardware* Bittau et al. (2017); Sasy et al. (2022), *multi-party computation* Lu et al. (2019); Abraham et al. (2020); Lu & Kate (2022), *MixNets* Dingledine et al. (2004); Chen et al. (2015); Langowski et al. (2023); Hohenberger et al. (2014); Sampigethaya & Poovendran (2007), which can also be verifiable Kwon et al. (2015; 2017) and *DC networks* Corrigan-Gibbs & Ford (2010); Corrigan-Gibbs et al. (2015). In other words, the shuffler does not necessarily need to be a single separate server; its functionality could be performed distributively for instance even by the clients themselves (cf. Section 4). This appealing trust assumption of the shuffle model has led to its adaptation from tech giants, such as Google Bittau et al. (2017) and Brave Minto et al. (2021).

In DP, the shuffle model was first introduced by Bittau et al. (2017) and later formalized in Cheu et al. (2019). This model has been applied in FL in a variety of ways to amplify the privacy by adding an additional layer of anonymity Sun et al. (2020); Xu et al. (2024); Girgis et al. (2021); Yang et al. (2023); Liu et al. (2021); Huang et al. (2023); Lebrun et al. (2022). Each user perturbs their model updates, then sends them to the shuffler which randomly permutates them before releasing them to the central server.

As far as we are aware of, no defense for SIAs has been studied in the literature, which is the motivation for this work. We have previously presented a preliminary shuffle-based approach to defending against SIAs in a poster Athanasiou et al. (2024), which partially reduced the accuracy of SIAs but was practically unfeasible due to its high communication cost.

## 3 PRELIMINARIES

**Source inference attacks (SIAs) Hu et al. (2021; 2024)**    The adversary (central server) knows that a training record $z = (x, y)$ (where $x$ is an input vector and $y$ is a class label) is present in the training dataset. The source status of each record can be represented by an $n$-th dimensional multinomial vector $s$ (where $n$ is the number of clients). Only one element of $s_{i,j}$ is equal to 1 (indicating that client $i$ owns the $j$-th record) and the others are set to 0. For the $j$-th record $z_j$, the source inference attack can be defined as follows:

**Definition 1** (Source inference attack). *Hu et al. (2021; 2024) Given a local optimized model $w_i$ and a training record $z_j$, source inference aims to infer the posterior probability of $z_j$ belonging to the client $i$:*

$$\mathcal{S}(w_i, z_j) := P(s_{i,j} = 1 \mid w_i, z_j)$$

**Encoding Schemes**    In this work, we combine shuffling with the following encoding schemes:

**Definition 2** (Unary encoding). *If $x, k \in \mathbb{N}, x \le k$, then $x$ is encoded as: $\mathcal{U}(x, k) = \{1\}^x \cup \{0\}^{k-x}$*

Moreover, we employ the residue number system (RNS) Garner (1959):

**Definition 3** (RNS encoding). *If $m_1, m_2, \ldots, m_u$ are pairwise coprime, then $x \in \mathbb{N}$ is encoded as $\{x_1, x_2, \ldots, x_u\}$ where $x_i = x \bmod m_i$.*

Given the residues $x_1, x_2, \ldots x_u$, one can reconstruct $x$ using the *Chinese remainder theorem* (we show an example in Appendix C.1). Also, Definition 3 can be extended for $x \in \mathbb{Z}$. If $x < 0$, the encoding is first performed for $|x|$, getting the resulting residues $x_1, x_2, \ldots x_u$. Then, $x$ is encoded as $\{m_1 - x_1, m_2 - x_2, \ldots, m_u - x_u\}$. Furthermore, the addition of two RNS encoded numbers can be performed directly in the RNS domain by summing their residues. Finally, the range of the RNS encoding is defined by the choice of the moduli:

**Proposition 3.1.** *$x \in \left[ -\lfloor \frac{M}{2} \rfloor, \lfloor \frac{M-1}{2} \rfloor \right] \cap \mathbb{Z}$ can be losslessly encoded in the RNS, where $M = \prod_i m_i$.*

## 4  SETTING, ASSUMPTIONS & THREAT MODEL

In this section we clarify the setting on which we will focus and define the attacker.

**Cross-Silo setting**  SIAs were proposed for the so-called *cross-silo* setting of FL, where the number of clients is limited (typically 2-100 clients Kairouz et al. (2019)) but each has superior communication/computation capabilities. For instance, it can be hospitals cooperating to produce a joint model for some disease. This model is widely applied in real-world scenarios, ranging from medical data Heyndrickx et al. (2024); Google Cloud (2023); Mateus et al. (2024); TNO (2025); Darzi et al. (2024); Dayan et al. (2021) to the agricultural domain Durrant et al. (2022); Dembani et al. (2025). The opposite setting, known as *cross-device*, involves a larger number of clients, each with limited communication and computational capabilities (e.g. smart watches and a health-monitoring model).

Studying SIAs for the cross-silo setting is more natural, as the attack relies on the fact that each client has a specific attribute linked to its data points. In other words, a sensitive attribute that the adversary infers upon identifying which client trained a given data point. For example, a hospital is typically associated with the medical conditions it specializes in, making it likely that its patients suffer from a respective disease. Therefore, in this work, we focus on the cross-silo setting, in line with previous research on SIAs Hu et al. (2021; 2024).

In previous works Hu et al. (2021; 2024), SIAs focused on supervised learning for classification tasks, as we do as well. We also assume that model parameters are clipped in $(-1, 1)$; this can be achieved by clipping and properly scaling them. Finally, we focus on the FedAvg aggregation function, but our approach can be generalized to any sum-based aggregation function (cf. Section 8).

**Shuffle Model**  We assume a shuffle model architecture (e.g. a mechanism from Lebrun et al. (2022); Sun et al. (2020); Xu et al. (2024); Girgis et al. (2021); Yang et al. (2023); Liu et al. (2021); Huang et al. (2023)) and our goal is to propose a defense mechanism which requires minimal modifications. A natural question arises: *why trusting a shuffler is more appealing than trusting the central server?* Shuffling is a primitive operation that can be performed distributively. Thus, the trust assumption can be distributed across different entities, whereas model aggregation (e.g. FedAvg) cannot (at least without heavy computational cost; we compare with Secure Aggregation in Section 6.2).

To highlight this, we give an example implementation with MixNets (Alg. 8). Our implementation can be adjusted to three levels of trust assumptions on the shuffler: *Fully Trusted* (allowed to receive plain-text secrets), *Semi-Honest* (follows the protocol but should not be able to see plain-text secrets) and *Partially Malicious* (all but one shuffle entities can deviate from the protocol arbitrarily; for example all but one servers of the MixNets may decide not to shuffle). For the *Partially Malicious* setting, Alg. 8 uses Onion Encryption Dingledine et al. (2004) (preventing the MixNet's servers from seeing the messages) and Zero-Knowledge Proofs (ZKPs) (preventing the servers from modifying the messages). In the *Semi-Honest* setting, ZKPs can be omitted since we assume that no server attempts to alter the data. In the *Fully Trusted* setting, neither Onion Encryption nor ZKPs are necessary.

In other words, Algorithm 8 requires that *only one* server of the MixNet be (semi or fully) trusted; all the others can be malicious. If we require each FL client to run their own server in the MixNet (a reasonable design choice given their resources in the cross-silo setting), then each client only has to trust themselves and no other entity, which satisfies our design specification **S.5**.

**Attacker**  We consider an honest-but-curious central server as an adversary who sees the output of a trusted shuffler (or at least partially trusted with MixNets, i.e. at least one server in the MixNet is assumed to be trusted). We assume that the shuffler does not collude with the adversary (e.g. using the aforementioned Algorithm 8). The attacker knows that a particular data point $z$ was used in the training phase (e.g. from a MIA) and their goal is to find, using an SIA, the client who owns it.

If no other information is known to the adversary then the standard shuffle model is sufficient to protect against SIAs since shuffling breaks the link between the data owner and their value. However, assuming that the adversary does not know anything more about the target client is arguably a strong assumption. In this work, we consider that the attacker knows a small *shadow dataset* only of the target client. A shadow dataset $S_x$ of a target user $x$ is a dataset disjoint from the one that $x$ actually uses for training, which follows (approximately) the same distribution. For instance, using once more the hospital example, if the attacker is a pharmaceutical company it might already know that some of

its patients were treated by the target hospital. Note that a similar assumption is often made in other privacy attacks like the MIA Shokri et al. (2017); Carlini et al. (2022); Choquette-Choo et al. (2021). We clarify that this shadow dataset will only be used to reverse the shuffling (Section 5); SIAs do not rely on the shadow dataset at all (Definition 1).

# 5 RECONSTRUCTION ATTACKS AGAINST SHUFFLING

In this section we show how an attacker can remap a shuffled model back to its original owner $x$ by using their shadow dataset $S_x$. The attacker here is the central server; in this section we assume no collision with the shuffler who is fully trusted. We begin with model-level shuffling, the standard approach used in the shuffle model of FL. Then, as illustrated in Figure 5, we expand the granularity to include layer-level and parameter-level shuffling, showing that in all three cases reconstruction attacks are possible. The algorithms discussed in the following subsections are placed in the Appendix B.

## 5.1 MODEL-LEVEL SHUFFLING

The current widely-used approach in shuffle-based FL is shuffling at the level of model updates. In other words, the shuffler receives the model $w_i$ from each user $i$, chooses a random permutation $\pi$ and outputs $w_{\pi(1)}, \ldots, w_{\pi(n)}$. However, the adversary can defeat the shuffler by remapping the model back to the target client $x$, by comparing the accuracy of each model on the shadow dataset $S_x$ (Algorithm 2). The model with the highest accuracy on $S_x$ will, most probably, belong to client $x$. Observe that this attack is feasible for the adversary; if $n$ is the number of clients and $||S_x||$ is the size of the shadow dataset, then Algorithm 2 has $O(n \cdot ||S_x||)$ complexity (as the adversary needs to find the accuracy on each of the samples of $S_x$ on each of the clients).

## 5.2 LAYER-LEVEL SHUFFLING

Since standard (model-level) shuffling can be reversed, let us explore shuffling per layer of the neural network Lebrun et al. (2022). For instance, if we consider a simple CNN with a convolutional layer $\mathcal{C}$ and 2 FC layers $FC1, FC2$, the shuffler releases: $\{\mathcal{C}_{\pi_0(1)}, \ldots, \mathcal{C}_{\pi_0(n)}, FC1_{\pi_1(1)}, \ldots, FC1_{\pi_1(n)}, FC2_{\pi_2(1)}, \ldots, FC2_{\pi_2(n)}\}$. In other words, for each FC and convolutional layer, the shuffler finds a *new* random permutation $\pi$. This does not affect the accuracy of the joint model because in FedAvg the order of the received layers is inconsequential. Observe now that the adversary has to run Algorithm 2 on every possible combination of layers. Depending however on the model, the number of combinations can be significant, which might make this approach non-feasible. To counter this, we propose a strategy which focuses only on the final layer of the neural network (Algorithm 3). Assuming w.l.o.g. that this is a FC layer, say $FC_L$, the adversary focuses on correctly remapping each $FC_L$ back to its original owner. For the remaining layers, the adversary computes their average (using FedAvg). This reduces the complexity again back to $O(n \cdot ||S_x||)$, for $n$ clients. The intuition behind this approach is that the final layer of a model tends to contribute more to overfitting. As a result, it more strongly reflects the distributional differences of the inputs. This is precisely what SIAs exploit, enabling the adversary to focus on the final layer, which provides the greatest advantage.

## 5.3 PARAMETER-LEVEL SHUFFLING

Next, let us consider shuffling per dimension, i.e. for each parameter $p$ of each layer the shuffler releases $p_{\pi(1)}, \ldots, p_{\pi(n)}$, an approach not yet explored in FL, as far as we are aware. Each random permutation $\pi$ has to be resampled for each parameter $p$ as otherwise the adversary could target the most easily distinguishable parameter (in case there are such outliers) to retrieve the permutation [1]. This approach does not affect the accuracy of the joint model; the central server can compute the aggregated model unimpeded. This creates an obvious obstacle to the adversary: finding the best accuracy of each possible combination of parameters requires superior computational capabilities. To overcome the hurdle, we follow a similar approach to Algorithm 3. That is, we once again focus only on reconstructing the final layer of the neural network (say $FC_L$ with $k$ parameters), taking the average of the other ones. However, now we first compute the average of $FC_L$ and change each

---

[1]We assume clients send all parameters at once, with the shuffler applying distinct permutations using sufficient metadata. Alternatively, parameters can be sent individually, which increases communication rounds.

of its parameters $p$ one by one. For each one, we select to keep the one out of the $n$ choices (one from each client) that creates a model with the best accuracy on $S_x$. To summarize, we copy the global model; each time we change only one of its parameters from the final layer, examining the possible choices and storing the one with the highest accuracy (Algorithm 5), which has a complexity of $O(k \cdot n \cdot ||S_x||)$.

## 6 PROTECTION AGAINST SOURCE INFERENCE ATTACKS

### 6.1 PARAMETER-LEVEL SHUFFLING WITH ENCODING

While parameter-level shuffling presents the greatest challenge for an attacker to reconstruct, the attack still remains feasible especially for heterogeneous datasets, as we show experimentally in Section 7. In this section, we argue that to effectively defend against SIAs, parameter-level shuffling should be combined with encoding, which further enhances the granularity of shuffling.

To achieve robust protection, we increase the shuffling granularity to bits. We show that this ensures that only the aggregated result of each parameter is leaked to the central server (which is necessary to compute the joint model) and not any information about the individual models. First, we compress the values using the residue number system (RNS) and then encode them into bits, using unary encoding (Definition 2). An outline of the proposed method is presented below; it is described in detail in Algorithm 1, with an example shown in Figure 4.

Recall from Section 3 that RNS encodes an integer $x$ by performing a modulo operation with several pairwise coprime integers (called *moduli*). Since RNS is designed for integers [2], we consider a scaling factor $10^r$, known to both the clients and the server, where the *precision* $r$ corresponds to the number of digits of the fractional part that will be transmitted. Thus, every parameter $p \in (-1, 1)$ becomes $\lfloor p \cdot 10^r \rceil$ and is then encoded using RNS. Each residue is then unary encoded and submitted in a separate shuffling round. The server can find the sum of each residue and, using the Chinese remainder theorem, reconstruct the sum of each parameter (which would need to be descaled by $10^r$).

---

**Algorithm 1:** `Parameter-level shuffling with RNS`

---

**Input:** $n$ clients, precision $r$, a function $RNS_{enc}(\cdot)$ for RNS encoding and $RNS_{dec}(\cdot)$ for decoding, a function $\mathcal{U}(x, k)$ for unary encoding $x$ in $k$ bits

**Output:** $w_{glob}$, the joint model

Let $\mathcal{M} = \{m_1, m_2, \ldots, m_u\}$ be pairwise coprime integers where $\prod_{m \in \mathcal{M}} m < n(10^r - 1)$

**Client-side (each client $i$ with model $w_i$):**

// Encode each parameter in RNS

**for** *each parameter $p$ in $w_i$* **do**

    $\{p_1, p_2, \ldots p_u\} \leftarrow RNS_{enc}(\lfloor p \cdot 10^r \rceil, \mathcal{M})$

    **for** *each residue $p_i$ in $\{p_1, p_2, \ldots p_u\}$* **do**

        Send $\mathcal{U}(p_i, m_i)$ to the shuffler.

**Shuffler:**

**for** *each parameter $p$* **do**

    Concatenate all the received bit vectors to a vector $B_{p,j}$ for each residue $j$.

    Shuffle (bit-wise) each $B_{p,j}$ and send it to the central server.

**Server-side:**

Receive the permuted bit vectors $B'_p$ for each parameter $p$.

**for** *each $i$-th parameter of $w_{glob}$* **do**

    // Sum and decode the residues to get the average of each parameter

    $y \leftarrow (\sum B'_{p,0}, \sum B'_{p,1}, \ldots, \sum B'_{p,u})_{RNS}$

    $w_{glob}[i] \leftarrow (RNS_{dec}(y, \mathcal{M})/10^r)/n$

**Return** $w_{glob}$

---

[2]Floating-point RNS Chiang & Lu (1991) is unsuitable as we cannot sum the values by their shuffled residues.

**Meeting the required specifications**  Our main insight is that revealing a shuffled bit vector is privacy-wise equivalent to revealing its sum (Proposition A.2). We show that Algorithm 1 releases only the aggregated model, without revealing any information about the individual local models. Since SIAs work by evaluating the accuracy of the given data point on the individual local models, this reduces their application to random guess, as there are no distinct models to compare.

Formally, we show that Algorithm 1 satisfies design specification **S.1** (Protection), in Theorem 1:

**Theorem 1.** *Algorithm 1 reduces the accuracy of SIAs to the level of random guessing.*

The proof can be found in Appendix A. Theorem 1 holds even when an adversary attempts to correlate information across epochs to improve their SIA success rate, since the proof shows that the only information leaked in each round is the sum of the models (which is useless for SIAs). Since no information about the individual models is revealed, inter-round correlation offers no advantage.

Then, **S.2** (Integrability) is also respected since only encoding/decoding operations need to be added. Regarding the communication cost (**S.3**), assuming $n$ clients, the moduli $m_1, m_2, \ldots, m_u$ should be picked such that $n \cdot v < \lfloor \frac{(M-1)}{2} \rfloor$ (Proposition 3.1) where $M = \prod_i m_i$ and $v = 10^r - 1$ (i.e. the largest integer with $r$ digits). Therefore, assuming $u$ moduli with $m_u$ being the largest, the cost is $O(u \cdot m_u)$. The number of shuffling rounds can be kept reasonably small, as it will be equal to the number of moduli (Figure 11). In general, small values are usually used for the residues ($m_i$), hence each user has to send a small number of $m_i$ bits in each round, keeping the communication cost reasonable, as we show in Section 7. Furthermore, **S.4** (Accuracy) is preserved with a sufficient $r$ since RNS encoding is lossless (Claim A.1). Finally, our proposed solution does not require additional trust assumptions than the ones of the shuffle model. Our proposed defense is fully compatible with the example implementation of shuffling with MixNets (Alg. 8), where each user only has to trust themselves, satisfying **S.5**.

In summary, the combination of RNS, unary encoding and parameter-level shuffling enables strong privacy without sacrificing much efficiency. We use Proposition A.2 to introduce a novel approach to privacy amplification without adding any noise. By combining it with RNS, we can minimize the bit vectors and reduce communication cost. With this approach, we provide a novel noise-free yet optimal privacy protection, along with compression to limit efficiency loss. Alg. 1 is compatible for all three defined trust assumptions on the shuffler *(Fully Trusted, Semi-Honest, Partially Malicious)*. For *Fully Trusted* and *Partially Malicious* we can optimize Algorithm 1 more, as we discuss below. Moreover, Algorithm 1 is fully compatible with DP-SGD as encoding and shuffling can be considered post-processing operations of model obfuscation. We further explore the synergy in Appendix F.8.

**Fully Trusted Shuffler**  Algorithm 1 does not rely on the shuffler to perform any communication optimizations. This allows for encrypted communication (e.g. Onion Encryption in Algorithm 8), preventing the shuffler from seeing the actual secrets. However, the typical assumption in the literature of the shuffle model is that the shuffler is fully trusted Girgis et al. (2021). In that case, encryption is not necessary and the communication can be further optimized by coupling Algorithm 1 with compression techniques such as Run-Length Encoding (RLE); we describe the corresponding algorithm in Appendix B.2, which we further evaluate in Section 7.

**Partially Malicious Shuffler**  Algorithm 1 can be coupled with Zero-Knowledge Proofs (ZKPs) to ensure that no MixNet server alters the data (as in Algorithm 8). Since ZKPs are costly, requiring one for every model parameter would lead to a severe computational overhead. Instead, a more practical approach is to select a subset of parameters to pair with ZKPs, effectively serving as "traps" for misbehaving servers. If a MixNet server tampers with the data of any such parameter, the corresponding ZKP will fail, flagging the server as malicious for further action (e.g. removal from the MixNet). These parameters may be selected at random, though one could improve by choosing parameters known to localize memorization more strongly Maini et al. (2023).

## 6.2  COMPARISON WITH SECURE AGGREGATION

One can view our approach as essentially turning the shuffler into a secure aggregator from the adversary's point of view by properly combining shuffling with encoding. A natural question thus arises about how our method compares with Secure Aggregation (SA) Bonawitz et al. (2017).

Applying threshold secret-sharing requires $t$ out of $n$ clients to reconstruct a secret, in order to minimize the cost and the risk of dropouts, where $t << n$ (e.g. $t = 0.5 \cdot n + 1$ Bonawitz et al. (2017)). Therefore, the protocol is private to any $t - 1$ malicious coalitions of clients. On the contrary, shuffling with MixNets requires only 1 out of $n$ servers to be trusted (e.g. Algorithm 8). Thus, the two models need to be compared under a common denominator, which is a common trust assumption. This can be achieved by implementing SA with standard threshold secret-sharing with $t = n - 1$, aligning with the trust assumption of the shuffle model (and our design specification **S.5.**). Each user must send $C$ bits to every other user, where $C$ is the size of each share, increasing the communication cost to $C \cdot (n - 1)$. Now the liveness constraints become a significant problem since, even if one client drops out the secret cannot be recovered. In contrast, MixNets can still shuffle and release the output even if all but one server drops out. In conclusion, SA incurs a stronger trust assumption and liveness constraints; we compare nevertheless our approach with this setting in Section 7.

# 7 EVALUATION

This section presents the primary experiments; additional evaluation can be found in Appendix F.

**Setting** For the SIAs we use the same code as Hu et al. (2021; 2024) in order to provide comparable results, which also includes the structures of the neural networks. We use a CNN for MNIST and CIFAR-10 and a ResNet-18 for CIFAR-100 (cf. Appendix D). The success rate of the attack is defined as the percentage of the correct guesses on the given data points. Training is performed across 20 rounds for MNIST and across 100 rounds for CIFAR-10 and CIFAR-100 but only the best SIA accuracy is reported. The number of local epochs is set to 10. As an upper baseline for the experiments we use the accuracy of the SIA when no shuffling is used (vanilla FL) and as a lower baseline the random guess (uniform over the clients). We split the dataset among the clients using the Dirichlet distribution with a parameter $\alpha$ (level of heterogeneity). Finally, we assume that the adversary has a shadow dataset of $5\%$ the size of the target client's actual training dataset. We also repeat the experiments for the $0.5\%$ and $1\%$ dataset sizes and evaluate the case where the adversary has access to a noisy shadow dataset. Both variations lead to similar conclusions and are therefore in Appendix F.2. Since the communication cost of Algorithm 1 depends on the sum of the RNS co-primes, we select the smallest possible pairwise co-primes that satisfy the constrain $n \cdot v < \lfloor \frac{(M-1)}{2} \rfloor$, where $v = 10^r - 1$ and $n$ is the number of clients (as explained in Section 6).

## 7.1 PROTECTION AGAINST SIAS

In this experiment, we test the SIA accuracy on the proposed Algorithm 1 and also on the Algorithms 2, 3 and 5 that remap the shuffled models back to their original owners from Section 5. Figure 1 shows the results relative to the level of heterogeneity ($\alpha$). We note that the results below do not depend on the shuffler's trust assumption, since in all cases we assume that the shuffler leaks no information (either through the use of Onion Encryption or by fully trusting the shuffler).

Model-level shuffling is the least effective choice; Algorithm 2 can reliably reverse shuffling, particularly when $\alpha$ is small. Notably, SIAs demonstrate a high success rate on CIFAR-100 with ResNet-18,

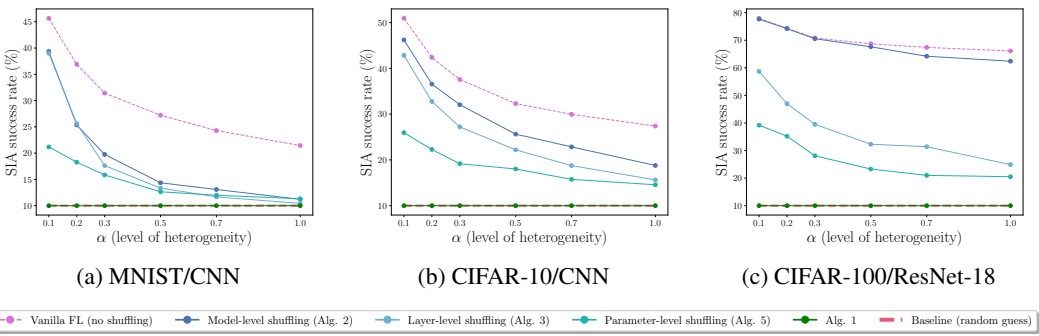

Figure 1: Success rate of SIA for 10 clients

as the increased complexity of the dataset amplifies the differences between the generated local models. Layer-level shuffling exhibits slightly lower performance on MNIST and CIFAR-10 than Algorithm 2. On CIFAR-100, however, the difference is larger; for instance, Algorithm 2 achieves approximately $\approx 78\%$ accuracy, while Algorithm 3 reaches only $\approx 60\%$ for $\alpha = 0.1$. This larger gap may be explained by the increased complexity of the ResNet architecture, where overfitting may also be stored in layers other than the final one, which Algorithm 3 targets. Parameter-level shuffling is the best choice among the three. Algorithm 5 cannot reconstruct as well the shuffled models; the accuracy of the SIA is reduced, for instance from $50\%$ to $28\%$ on the CIFAR-10 for $\alpha = 0.1$. Still, however, the SIA accuracy remains higher than random guessing. Our proposed solution of Algorithm 1 always reduces the accuracy to the level of random guess, validating Theorem 1. In conclusion, the experiment shows that a straightforward application of shuffling might not be enough to protect against SIAs, especially in scenarios with high heterogeneity which are arguably the most relevant cases for SIAs. In contrast, our proposed solution provides an optimal defense. We also evaluate with more clients and other aggregation functions, such as FedSGD and FedProx, as well as an MLP network, which are placed in Appendix F, since they lead to similar conclusions.

**Protection against DRAs**   Although our main focus in this paper is on defending against SIAs, we also empirically demonstrate in Appendix F.3 that Algorithm 1 can also mitigate Data Reconstruction Attacks (DRAs). DRAs have been shown to depend heavily on the training batch size Yin et al. (2021). By applying Algorithm 1, we shuffle the models at such a fine granularity that only the aggregated model is exposed, effectively simulating a larger batch size. For example, if the batch size is 1 with $n$ clients, Algorithm 1 yields an aggregated model similar to having a single client train with a batch size of $n$. Specifically, while the reconstruction loss of the original DRA of Zhu et al. (2019b) is $3 \cdot 10^{-4}$, it increases to 0.98 when Algorithm 1 is applied with 10 clients on the CIFAR-10 dataset. Consequently, the reconstructed images become heavily degraded, containing no recognizable features. We elaborate further in Appendix F.3, showing additional experiments.

## 7.2   PERFORMANCE ANALYSIS

**Accuracy of the joint model**   Algorithm 1 only considers the first $r$ digits of the fractional part of each parameter. In this experiment, we show that $r$ can be kept reasonably small to minimize the communication cost while preserving the accuracy of the joint model. Figure 3 shows that for the MNIST, which requires a simpler model, $r = 2$ suffices to reach a level of accuracy comparable to vanilla FL and $r = 3$ scores a similar accuracy. On the other hand, CIFAR-10 and CIFAR-100, which require more complex models, need $r = 3$ digits to approach the performance of vanilla FL, and $r = 8$ to match it. Note that the results hold for all three trust assumptions on the shuffler, as we assume shuffling does not degrade model accuracy (either with trust or with ZKPs).

**Communication Cost**   In Figure 2, we compare the communication cost of our approach against the baseline of vanilla FL (standard 32-bit binary encoding), which however transmits the whole value (whereas Algorithm 1 transmits only the first $r$ digits). As an upper baseline we compare our approach with Secure Aggregation (SA), implemented with standard secret-sharing, with $t = n - 1$ (cf. Section 6.2), setting the smallest possible $C$ to avoid overflows (for the first $r$ digits). Moreover, we evaluate the combination of Algorithm 1 with RLE compression which further reduces the communication cost, when the shuffler is *Fully Trusted* (described in Appendix B.2). We do not compare SA with the *Partially Malicious* case, as SA assumes trusted or semi-honest clients. The *Partially Malicious* setting incurs the same communication cost as the *Semi-Honest* case for parameters not using ZKPs, plus an extra cost for those that do, depending on the chosen ZKP.

For 10 clients and $r$=5 the expansion factor, compared with vanilla FL, is $1.81\times$ on CIFAR-100/ResNet (Figure 14), which yields $\approx 53\%$ accuracy (compared to $\approx 55\%$ vanilla accuracy). This is an arguably manageable additional cost for the cross-silo setting, considering that the state-of-the-art protocol for SA for the cross-device setting exhibits an expansion factor of at least $1.73\times$ Bonawitz et al. (2017). Note that a direct comparison between the two is not meaningful as Bonawitz et al. (2017) relies on a stronger trust assumption (Section 6.2). Nevertheless, if an expansion of $1.73\times$ is deemed acceptable for the cross-device setting, a slightly higher factor should also be considered reasonable for the cross-silo setting, given the significantly superior communication capabilities of the clients. Finally, if we follow the literature of shuffle-model FL and assume a *Fully Trusted shuffler*, then the expansion factor drops to $1.03\times$, approaching the cost of vanilla FL (Figure 14).

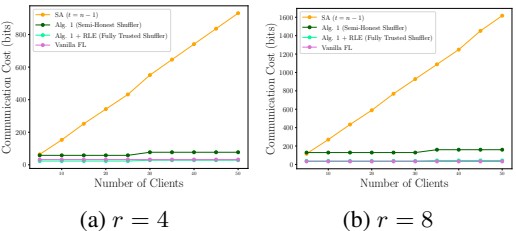

(a) $r = 4$        (b) $r = 8$

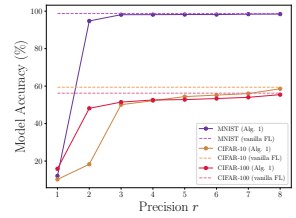

Figure 2: Communication Cost (bits per client per parameter).

Figure 3: Model Accuracy (top-1) of Alg. 1 with precision $r$ compared to vanilla FL (10 clients).

Table 1: Summary: Communication Cost for negligible (<3%) accuracy loss (*Semi-Honest* shuffler)

| Model | Dataset | Type | $r$ | Expansion |
|-------|---------|------|-----|-----------|
| CNN | MNIST | Image | 3 | 1.04× |
| CNN | CIFAR-10 | Image | 5 | 1.81× |
| ResNet | CIFAR-100 | Image | 5 | 1.81× |
| MLP | Synthetic Hu et al. (2024) | Tabular | 4 | 1.28× |
| CNN | HAR Reyes-Ortiz et al. (2012) | Time-Series | 4 | 1.28× |
| Transformer | Newsgroups Lang (1995) | Text | 4 | 1.28× |

Table 9 shows that Algorithm 1 scales well as the number of clients and $r$ increase. Even with $10^4$ clients (rather unrealistic in the cross-silo setting) and $r = 16$, Algorithm 1 needs 440 bits per parameter (or 79 bits with compression under a fully trusted shuffler), compared to 32 bits for vanilla FL and $\approx 83$ KBs for SA. Experiments with more complex models and datasets (Table 1) show that $r = 4$ suffices for optimal SIA protection with negligible accuracy loss (cf. Appendix F.2.2). The computation cost of Alg. 1 for encoding is negligible as it is based on primitive operations. For instance, the 11 million parameters of ResNet can be encoded in just 19 seconds (cf. Appendix F.6).

## 8 CONCLUSION

We explored defenses against SIAs in FL by combining shuffling with encoding. Our proposed mechanism is easily integrable within the shuffle model of FL and fully compatible with other privacy protection mechanisms, such as DP, as its addition does not affect DP guarantees or further degrade accuracy (cf. F.8). Another contribution of our work, potentially of independent interest, is the model reconstruction attacks to defeat a standard shuffler in datasets with a high level of dissimilarity. Future work should evaluate this attack on the state-of-the-art FL mechanisms of the shuffle model, as it has been shown that different mechanisms provide varying privacy guarantees when the shuffler is compromised Athanasiou et al. (2025). In this work, we focused on the shuffle model of FL. Developing similar defenses in settings where there is no shuffler remains an interesting step for future research. Moreover, we focused on FedAvg but our approach can be generalized to any other sum-based aggregation function, such as Li et al. (2020); Reddi et al. (2020); Li & Wang (2019); Ghosh et al. (2020), where the central server sums the parameters and then performs further processing. To verify, we performed experiments on FedSGD and FedProx (cf. F.4). Our work covers the vital class of sum-based aggregation functions which is widely used, for example in medical applications Mateus et al. (2024); TNO (2025); Darzi et al. (2024); Dayan et al. (2021). On the other hand, our approach is not directly applicable to non-sum-based frameworks, such as median-based, clustering-based and ranking-based (e.g. Yin et al. (2018); Ghosh et al. (2020); Iqbal et al. (2025)). We include a preliminary discussion and evaluation of applying our proposed method to such cases in Appendix F.9, which we aim to explore further. We did not consider disaggregation attacks Marchand et al. (2023), where the server observes only the joint model and attempts to recover local models. Combining them with SIAs is an interesting direction for future work, Finally, since our technique (Alg. 1) reveals only the sum of the secrets, it can serve as a secure aggregation protocol. It would be interesting to explore other use cases where revealing only the aggregated model suffices for protection, such as DRAs, which we evaluated in this work.

ACKNOWLEDGMENTS

This research was supported by the EU project ELSA: *European Lighthouse on Secure and Safe AI* (EU grant agreement 101070617) and by the PREP Cybersecurité project IPoP. The work of Andreas Athanasiou was also supported by the European Commission under the Horizon Europe Programme as part of the project RECITALS: *An open-source platform for Resilient sECure digITAL identitieS* (EU grant agreement 101168490).

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

## SUPPLEMENTARY MATERIAL

**Ethics statement**   As discussed in Section 1, SIAs can pose a significant real-world privacy risk. Our proposed defense offers robust protection, making a positive contribution to user privacy. Notably, its ease of integration allows clients to implement the defense promptly. On the other hand, the proposed reconstruction attacks of Section 5 can undermine the privacy guarantees of shuffle-based FL mechanisms, thereby compromising the privacy of end users. Additional work is needed to verify their effectiveness across other shuffle-based mechanisms.

**Reproducibility statement**   To make integration easier with current shuffle model mechanisms, we have implemented Algorithm 1 and placed its code in a public repository [3]. Using this same code, all experiments reported in the main body and appendix can be reproduced; a corresponding README file is included in the repository as guidance.

**Use of LLMs**   We used Grammarly to fix awkward phrasing, grammatical errors and to improve the overall clarity of sentences.

---

[3]https://github.com/andathan/sia_defence

APPENDIX

## A  PROOFS

In this section we include the missing proof of Theorem 1. We will use the notion of *Markov chain* in the information-theoretic sense Cover & Thomas (2006). We recall that three random variables $X, Y$ and $Z$ form a Markov chain, denoted by $X \to Y \to Z$, if $X$ and $Z$ are conditionally independent given $Y$. Formally, $\forall x, y, z, \; P[Z = z | X = x, Y = y] = P[Z = z | Y = y]$, or, equivalently, $\forall x, y, z, \; P[X = x | Z = z, Y = y] = P[X = x | Y = y]$ where $P$ stands for probability. Note that if $X \to Y \to Z$, then also $Z \to Y \to X$.

We will also use other notions from information theory, namely *Shannon entropy* $H(X)$ and *conditional entropy* $H(X|Y)$, where $X$ and $Y$ are random variables. We recall that $H(X)$ represents the uncertainty about the value of $X$, and $H(X|Y)$ represents the remaining uncertainty about $X$ after we learn $Y$.

First, we recall the well-known principle in information theory known as the *data processing inequality* (DPI), which states that *postprocessing* cannot increase the amount of information.

**Proposition A.1.** *[Data processing inequality Cover & Thomas (2006)] Let $X \to Y \to Z$ be a Markov chain. Then the Shannon conditional mutual entropy $H(\cdot|\cdot)$ satisfies the following property:*

$$H(X|Y) \leq H(X|Z).$$

Since $H(X|Y)$ represents the remaining uncertainty about $X$ after we learn $Y$, the above formula $H(X|Y) \leq H(X|Z)$ means that the amount of information that $Y$ carries about $X$ is less than the amount of information that $Z$ (the result of postprocessing $Y$) carries about $X$.

To prove Theorem 1, we first prove that releasing a shuffled unary vector carries the same amount of information about the initial vector as releasing its sum:

**Proposition A.2.** *Let the random variable $V$ range over vectors of bits of fixed length $k$, namely over $\{0, 1\}^k$, let $S(V)$ represent the shuffling of $V$, and let $\Sigma V$ represent the sum of all elements of $V$. Assume that the shuffler is a probabilistic function that depends only on the number of 1's in the vector. Then, the information that $S(V)$ carries about $V$ is the same as the information that $\Sigma V$ carries about $V$. Namely:*

$$H(V|S(V)) = H(V|\Sigma V).$$

*Proof.* Since the sum of the elements of $V$ is the same as the sum of the elements of any shuffling of $V$, we have that $V$, $S(V)$ and $\Sigma V$ form a Markov chain. Namely:

$$V \to S(V) \to \Sigma V$$

Therefore, by Proposition A.1, we have:

$$H(V|S(V)) \leq H(V|\Sigma V) \tag{1}$$

On the other hand, since we are assuming that the result of the shuffling of $V$ (i.e., the probability distribution over the shuffled vectors) depends only on the number of 1's in $V$, we have that also $V$, $\Sigma V$ and $S(V)$ form a Markov chain:

$$V \to \Sigma V \to S(V)$$

From which we derive, applying again Theorem A.1, that

$$H(V|\Sigma V) \leq H(V|S(V)). \tag{2}$$

Finally, from Equations (1) and (2), we can conclude. $\square$

We note that a related result, formulated in terms of probabilities, was shown in Cheu et al. (2019); Athanasiou et al. (2025) in the context of differential privacy.

Let us now prove the following property of Markov chains:

**Proposition A.3.** *Consider the following Markov chains:*

$$X \to Y \to Z \,, \tag{3}$$

$$X \to Y \to W \,. \tag{4}$$

*If $H(Y|Z) = H(Y|W)$, then:*

$$H(X|Z) = H(X|W).$$

*Proof.* From the first Markov chain in Equation (3), we deduce:

$$
\begin{aligned}
H(X|Z) &= H(X,Y|Z) &&[X \to Y \text{ is deterministic}] \\
&= H(X|Y,Z) + H(Y|Z) &&[\text{Chain rule}] \\
&= H(X|Y) + H(Y|Z) &&[\text{Markov chain definition}]
\end{aligned}
$$

By applying the same reasoning to the second Markov chain in Equation (4), we obtain:

$$H(X|W) = H(X|Y) + H(Y|W).$$

Since $H(Y|Z) = H(Y|W)$, we conclude that $H(X|Z) = H(X|W)$. $\qquad\square$

Now we are ready to prove our main theorem about the protection of our mechanism.

**Theorem 1.** *Algorithm 1 reduces the accuracy of SIAs to the level of random guessing.*

*Proof.* We will show that the result of Algorithm 1 reveals no more information than the aggregated result (i.e. the joint model), thereby reducing the SIA accuracy to that of a random guess by definition, as there are no distinct models to compare. We assume $r$ is large enough so that the whole value is transmitted, which is the setting which leaks the most information.

Let $X^j$ be the random variable ranging over the collection of values that the users hold for some parameter $j$. Define for each residue $m_u$

$$\mathcal{Y}_u^j := \{x \bmod m_u : x \in X^j\}$$

and let $Y_u^j$ be the corresponding random variable. The shuffler in Algorithm 1 receives the unary encoded values, concatenates them to a single vector, and shuffles them altogether, outputting a shuffled version $S(\mathcal{Z}_u^j)$ of $\mathcal{Z}_u^j$, where

$$\mathcal{Z}_u^j := \bigcup_{y \in \mathcal{Y}_u^j} \mathcal{U}(y, m_u)$$

Namely, $\mathcal{Z}_u^j$ is the concatenation of the unary encodings of the elements of $\mathcal{Y}_u^j$. Note that $Y$ depends only on $Y_u^j$. The correlation between $X^j$, $Y_u^j$, and $S(\mathcal{Z}_u^j)$ (the latter being the vector observed by the central server), can be described by the following Markov chain:

$$X^j \to Y_u^j \to S(\mathcal{Z}_u^j) \tag{5}$$

Now, let us consider a scenario where the central server observes the aggregation (i.e., the summation) of all the values in $Y_u^j$, which we denote by $\Sigma Y_u^j$, rather than the shuffled version of the concatenation of their encodings. We have the following Markov chain:

$$X^j \to Y_u^j \to \Sigma Y_u^j \tag{6}$$

Since the summation of the elements of $Y_u^j$ is the same as the summation of their concatenation $Y$, we can rewrite Equation 6 as:

$$X^j \to Y_u^j \to \Sigma \mathcal{Z}_u^j \tag{7}$$

From Equation 5 and Equation 7 we derive, by Proposition A.2:

$$H(Y_u^j|S(Y)) = H(Y_u^j|\Sigma Y).$$

Given the above equality, and Equations (5) and (7), we can apply Proposition A.3 and conclude that, for each parameter $j$ and each residue $m_u$:

$$H\big(X^j|S(\mathcal{Z}_u^j)\big) = H\big(X^j|\Sigma \mathcal{Z}_u^j\big).$$

$\qquad\square$

The proof of Theorem 1 relies on the information that the central server first observes, since post-processing cannot increase the amount of information (*data processing inequality*). However, recall that Algorithm 1 instructs the server to decode the result. Therefore, the server can still recover the aggregated model using RNS addition (Section 3). The only potential loss of information may arise from the parameter $r$ (precision i.e., the digits transmitted after the decimal point). Since RNS encoding is lossless, we arrive at the following claim:

**Claim A.1.** *Assuming a sufficiently large $r$, Algorithm 1 does not impact the accuracy of the joint model.*

# B   ALGORITHMS

## B.1   RECONSTRUCTION ATTACKS OF SECTION 5

We begin with Algorithm 2 which reverses model-level shuffling by remapping the models back to their original owners using the accuracy on the shadow dataset $S_x$ of the target client $x$.

Then, for defeating layer-based shuffling, we assume w.l.o.g. 2 convolutional layers and 2 FC layers, presenting Algorithm 3. Note that for notational convenience, we use the FedAvg function to average layers (of neural networks) rather than entire models; the definition of the function remains similar.

Finally, Algorithm 5 shows a possible implementation of remapping parameter-level shuffling, assuming again w.l.o.g. 2 convolutional layers and 2 FC layers. Note that $\mathcal{FC}_{2,0}$ notates the shuffled set of the first parameters of the final FC layer $\mathcal{FC}_2$. Recall that each shuffle set has size $n$, if $n$ is the number of clients. We assume that $\mathcal{FC}_2$ has $k$ parameters.

---

**Algorithm 2:** $\mathcal{R}_\mathcal{M}$
Reconstruction attack against Model-level shuffling

---

**Input**  : $w'_1, \ldots, w'_n$, randomly permuted clients' models,
$\quad\quad\quad S_x$ shadow dataset of target user $x$,
$\quad\quad\quad A(w, D)$, accuracy of model $w$ on the dataset $D$
**Output**: $w_x$, the model of user $x$
$best \leftarrow w'_1$
**for** *each model $w_j$ in $w'_2, \ldots, w'_n$* **do**
$\quad$ **if** $A(w_j, S_x) > A(best, S_x)$ **then**
$\quad\quad$ $best \leftarrow w_j$
**Return** $best$

---

**Algorithm 3:** $\mathcal{R}_\mathcal{L}$
Reconstruction attack against Layer-level shuffling

---

**Input**  : $\mathcal{C}_1, \mathcal{C}_2$, randomly permuted convolutional layers, $\mathcal{FC}_1, \mathcal{FC}_2$, randomly permuted FC layers,
$\quad\quad\quad S_x$, shadow dataset of target user $x$,
$\quad\quad\quad A(w, D)$, accuracy of model $w$ on the dataset $D$
**Output**: $w_x$, a model with $FC_2$ of user $x$ and the average of the rest layers
`// For FC2 find the one with best accuracy; take the FedAvg for`
`   the rest`
**for** *each layer $L_i$ in $\mathcal{FC}_2$* **do**
$\quad$ $w \leftarrow CON(FedAvg(\mathcal{C}_1), FedAvg(\mathcal{C}_2), FedAvg(\mathcal{FC}_1), L_i)$
$\quad$ Append $w$ in $w_{models}$
**Return** $\mathcal{R}_\mathcal{M}(w_{models}, S_x, A)$

---

---

**Algorithm 4:** $CON$

Construct a model with given layers (used in Alg. 3)

---

**Input** : $C_1, C_2, FC_1, FC_2$ , layers of the model
**Output:** $w$, a model with the given layers
$w_{C1} \leftarrow C_1$
$w_{C2} \leftarrow C_2$
$w_{FC1} \leftarrow FC_1$
$w_{FC2} \leftarrow FC_2$
**Return** $w$

---

---

**Algorithm 5:** $\mathcal{R}_\mathcal{P}$

Reconstruction attack against Parameter-level shuffling

---

**Input** : $w_{glob}$ , the global model
$\mathcal{FC}_{2,0}, \ldots, \mathcal{FC}_{2,k}$, randomly permutated parameters of the final layer
$S_x$, shadow dataset of target user $x$,
$A(w, D)$, accuracy of model $w$ on the dataset $D$
**Output:** $w_x$, a model with $FC_2$ of user $x$ and the average of the rest layers
$w_x \leftarrow w_{glob}$
**for** *each $i$-th set of parameters $z_i$ in $\mathcal{FC}_{2,i}$* **do**
$\quad w_{best} \leftarrow REP(w_{glob}, i, z_{i,0})$
$\quad best \leftarrow z_{i,0}$
$\quad$ **for** *each parameter $j$ in $z_i$* **do**
$\quad\quad$ **if** $A(REP(w_{glob}, i, j), S_x) > A(w_{best}, S_x)$ **then**
$\quad\quad\quad w_{best} \leftarrow REP(w_{glob}, i, j)$
$\quad\quad\quad best \leftarrow j$
$\quad w_x \leftarrow REP(w_x, i, best)$
**Return** $w_x$

---

---

**Algorithm 6:** $REP$

Replace the $i$-th parameter of $FC_2$ with $p$ (used in Alg. 5)

---

**Input** : $w$, a model
$i$, the $i - th$ parameter of $FC_2$
$p$, parameter to replace
**Output:** $w'$, the output model
$w' \leftarrow w$
$w'_{FC_2}[i] \leftarrow p$
**Return** $w'$

---

### B.2 COMMUNICATION IMPROVEMENT FROM SECTION 6

If we assume that the shuffler is *Fully Trusted*, following the literature Girgis et al. (2021), then Onion Encryption is not necessary to hide the secrets (i.e. parameters) from the MixNet servers. Similarly, Zero-Knowledge Proofs are not required to verify that each server did not tamper with the data. In both cases, since the shuffler is honest, we assume it will follow the protocol and will not leak the secret values to an adversary. Thus, we can further improve the communication cost by adding a compression step after the unary encoding.

Run-Length Encoding (RLE) compresses a string by replacing consecutive repeated values with just a single value and a count of its repetitions. For example, AABAA will be represented as (2A,1B,2A). In our case we have bits; more specifically Unary Encoding (Definition 2) creates a bit-string by first placing the ones and then appending the zeroes. Since the size of the bit-string is known (i.e. $r$) we can send only the number of ones. For example, if we have $[1, 1, 1, 0, 0, 0, 0, 0, 0, 0]$ we can just send the number 3. The shuffler can create a bit-string with 3 ones and $r - 3 = 7$ zeroes. Algorithm 7 illustrates this idea.

---

**Algorithm 7:** `Parameter-level shuffling with RNS combined with RLE`

---

**Input:** $n$ clients, precision $r$, a function $RNS_{enc}(\cdot)$ for RNS encoding and $RNS_{dec}(\cdot)$ for decoding, a function $\mathcal{U}(x,k)$ for unary encoding $x$ in $k$ bits

**Output:** $w_{glob}$, the joint model

Let $\mathcal{M} = \{m_1, m_2, \ldots, m_u\}$ be pairwise coprime integers where $\prod_{m \in \mathcal{M}} m < n(10^r - 1)$

**Client-side (each client $i$ with model $w_i$):**

// Encode each parameter in RNS

**for** *each parameter $p$ in $w_i$* **do**

$\quad \{p_1, p_2, \ldots p_u\} \leftarrow RNS_{enc}(\lfloor p \cdot 10^r \rfloor, \mathcal{M})$

$\quad$ **for** *each residue $p_i$ in $\{p_1, p_2, \ldots p_u\}$* **do**

$\quad\quad$ Send $p_i$ to the shuffler. // Send the number of ones to the shuffler

**Shuffler:**

**for** *each parameter $p$* **do**

$\quad \mathcal{B} \leftarrow \mathcal{U}(p_i, m_i)$ for each RNS moduli $m_i$ and each received residue $p_i$. // Decompress

$\quad$ Concatenate all the decompressed received bit vectors $\mathcal{B}$ to a vector $B_{p,j}$ for each residue $j$.

$\quad$ Shuffle (bit-wise) each $B_{p,j}$ and send it to the central server.

**Server-side:**

Receive the permutated bit vectors $B'_p$ for each parameter $p$.

**for** *each $i$-th parameter of $w_{glob}$* **do**

$\quad$ // Sum and decode the residues to get the average of each parameter

$\quad y \leftarrow (\sum B'_{p,0}, \sum B'_{p,1}, \ldots, \sum B'_{p,u})_{RNS}$

$\quad w_{glob}[i] \leftarrow (RNS_{dec}(y, \mathcal{M})/10^r)/n$

**Return** $w_{glob}$

---

## C  EXAMPLE IMPLEMENTATIONS

This section discusses an example of the Chinese remainder theorem and gives an example implementation of shuffling with MixNets. It also includes the figures that are missing from the main-body due to space constraints. Figure 5 shows the different shuffling granularities which we examined in Section 5 and Figure 4 illustrates a simple example of Algorithm 1.

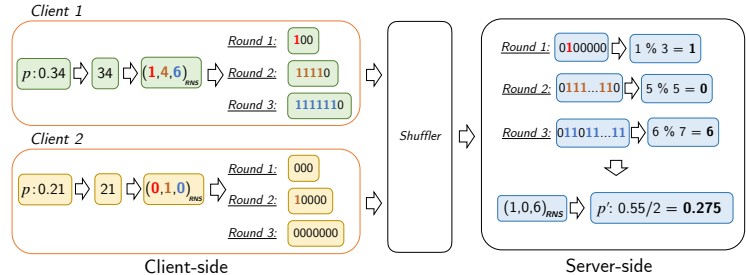

Figure 4: Example of Algorithm 1 with moduli $3, 5, 7$, for a parameter $p$, where $r = 2$ and $n = 2$.

### C.1  EXAMPLE OF THE CHINESE REMAINDER THEOREM

We show how the Chinese remainder theorem can be applied to reconstruct the secret value from the residues, using the example of Figure 4.

Consider the following system of congruences, where $3, 5,$ and $7$ are pairwise coprime:

$$\begin{cases} x \equiv 1 \bmod 3 \\ x \equiv 0 \bmod 5 \\ x \equiv 6 \bmod 7 \end{cases}$$

We begin by computing the product of the moduli:

$$M = 3 \times 5 \times 7 = 105.$$

For each congruence, we define $M_i = M/m_i$, where $m_i$ is the modulus of the $i$-th equation:

$$M_1 = 105/3 = 35, \quad M_2 = 105/5 = 21, \quad M_3 = 105/7 = 15.$$

Next, we compute the modular inverse $y_i$ such that $M_i \cdot y_i \equiv 1 \bmod m_i$:

$$35 \cdot y_1 \equiv 1 \bmod 3 \Rightarrow y_1 = 2,$$
$$21 \cdot y_2 \equiv 1 \bmod 5 \Rightarrow y_2 = 1$$
$$15 \cdot y_3 \equiv 1 \bmod 7 \Rightarrow y_3 = 1.$$

We now apply the Chinese remainder theorem formula:

$$x \equiv a_1 M_1 y_1 + a_2 M_2 y_2 + a_3 M_3 y_3 \bmod M,$$

where $a_1 = 1, a_2 = 0, a_3 = 6$.

By substituting the values we have:

$$x \equiv 1 \cdot 35 \cdot 2 + 0 \cdot 21 \cdot 1 + 6 \cdot 15 \cdot 1 = 70 + 0 + 90 = 160 \bmod 105.$$

Thus, the solution is:

$$x \equiv 55 \bmod 105.$$

Observe that $x = 55$ indeed satisfies all original congruences:

$$55 \bmod 3 = 1, \quad 55 \bmod 5 = 0, \quad 55 \bmod 7 = 6.$$

### C.2 SHUFFLING

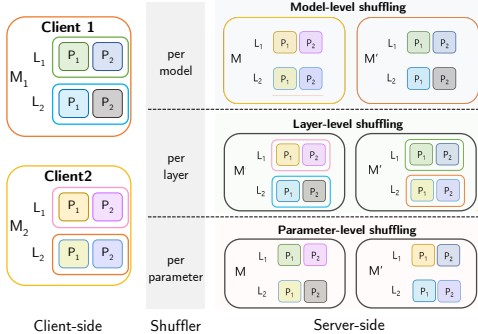

Figure 5: Shuffling methods in FL with different granularity

Algorithm 8 uses Onion Encryption Dingledine et al. (2004) to ensure that no server can view the messages and Zero-Knowledge Proofs (ZKPs) De Santis & Persiano (1992) to assert that no server can change the messages. Each server decrypts the outermost layer of onion encryption, verifies that no tampering has occurred with a ZKP, shuffles and sends the shuffled vector to the next server. Since an honest shuffling (i.e. the fact that the previous server chose a uniformly random permutation) cannot be proved via ZKPs, at least one server of the MixNet needs to be trusted.

In scenarios where the clients cannot run their own MixNet server, they can instead choose more servers from the network to shuffle. For instance, they can decide to always select 100 servers out of all the available ones (a different set of servers for each parameter), meaning that the adversary must control all 100 servers to compromise the defense for each parameter. However, this requires an omnipotent adversary that controls a significant part of the MixNet network (even hacking honest servers like those run by the honest clients). This attacker is arguably unrealistic; often, anonymity systems, like Tor Dingledine et al. (2004), assume that an attacker can only control parts of the network.

---

**Algorithm 8:** Shuffling with MixNet

---

**Input** : $M_i$: Message client $i$ wishes to send, $n$ number of clients, $P$: List of participating mix
servers, $\text{Onion}_{\text{Enc}}(M, \text{pk})$ and $\text{Onion}_{\text{Dec}}(M, \text{sk})$ for onion encryption/decryption
**Output** : Shuffled messages $M'$

```
// client-side
```
**for** *each client $i$ in parallel* **do**
$\quad C_i \leftarrow M_i$
```
   // onion-encryption
```
$\quad$ **for** *each mix server $j$ in reverse order of $P$* **do**
$\quad\quad\quad C_i \leftarrow \text{Onion}_{\text{Enc}}(C_i, \text{pk}_j)$
$\quad$ Send $C_i$ to $P_1$
```
// MixNet-side
```
**for** *each mix server $j$ in $P$* **do**
```
   // First server receives the messages (no ZKP to verify)
```
$\quad$ **if** $j = 1$ **then**
$\quad\quad\quad \mathbf{C} = [C_i]_{i=1}^n$
$\quad\quad\quad \pi \leftarrow \text{None}$
```
   // Last server outputs the messages
```
$\quad$ **if** $j = |P|$ **then**
$\quad\quad\quad$ **Return** MixVerify($\mathbf{C}$, $j$, $\pi$).
$\quad \mathbf{C}, \pi \leftarrow \text{MixVerify}(\mathbf{C}, j, \pi)$

---

**Algorithm 9:** MixVerify($C$,$j$,$\pi$)

---

**Input** : $\mathbf{C}$: Message from previous server, $j$: Current mix server, $\pi_{j-1}$ ZKP from previous
server, $\text{Verify}_{\text{ZKP}}(C, \pi)$ and $\text{Generate}_{\text{ZKP}}(\mathbf{C}, \mathbf{C}')$ to verify and generate ZKPs.
**Output** : Partially decrypted and shuffled message $\mathbf{C}'$

$M \leftarrow \text{Onion}_{\text{Dec}}(\mathbf{C}, \text{sk}_j)$
**if** $j > 1$ *and* $\neg$ *$\text{Verify}_{\text{ZKP}}(M, \pi_{j-1})$* **then**
$\quad$ **reject and abort**
$M' \leftarrow \text{Shuffle}(M)$
$\pi_j \leftarrow \text{Generate}_{\text{ZKP}}(M, M')$
**Return** $M', \pi_j$

---

## D  MODELS AND HYPERPARAMETERS

This section describes the models which we used in the experiments of Section 7. For MNIST and CIFAR-10 we used exactly the same models as in Hu et al. (2021; 2024) [4], as we used their code for evaluating the SIAs. For CIFAR-100 we used the ResNet-18, in order to illustrate that the proposed mechanism works even on more complex models.

### D.1  MNIST

We use a CNN which consists of two convolutional layers with 5×5 kernels, followed by two max-pooling layers, and three fully connected layers. Specifically, the architecture is as follows:

- *Conv2D layer*: 32 filters with a kernel size of $5 \times 5$. Activation: *ReLU*. Input shape: *(1, 28, 28)*.

- *MaxPooling2D((2, 2))*.

- *Conv2D layer*: 64 filters with a kernel size of $5 \times 5$. Activation: *ReLU*.

- *MaxPooling2D((2, 2))*.

- *Flatten layer*: Flattens the feature maps into a 1D tensor of size $64 \times 4 \times 4$.

---

[4]https://github.com/HongshengHu/SIAs-Beyond_MIAs_in_Federated_Learning

- *Dense layer*: 512 neurons. Activation: *ReLU*.
- *Dense layer*: 128 neurons. Activation: *ReLU*.
- *Output: Dense layer*: *1*0 neurons. Activation: *none (logits)*.

## D.2 CIFAR-10

The CNN consists of two convolutional layers with 5×5 kernels (the first convolutional layer takes input images with three color channels (RGB)), followed by two max-pooling layers, and three fully connected layers. Specifically, the architecture is as follows:

- *Conv2D layer*: 32 filters with a kernel size of $5 \times 5$. Activation: *ReLU*. Input shape: *(3, 32, 32)*.
- *MaxPooling2D((2, 2))*.
- *Conv2D layer*: 64 filters with a kernel size of $5 \times 5$. Activation: *ReLU*.
- *MaxPooling2D((2, 2))*.
- *Flatten layer*: Flattens the feature maps into a 1D tensor of size $64 \times 5 \times 5$.
- *Dense layer*: 512 neurons. Activation: *ReLU*.
- *Dense layer*: 128 neurons. Activation: *ReLU*.
- *Output: Dense layer*: *1*0 neurons. Activation: *none (logits)*.

## D.3 CIFAR-100

We use a standard ResNet-18 architecture He et al. (2016) for CIFAR-100. Specifically, we use the implementation provided by the PyTorch library, initialized without pretrained weights (pretrained=False). The architecture is as follows:

- *Conv2D layer*: 64 filters with a kernel size of $7 \times 7$, stride 2, padding 3. Activation: *ReLU*. Input shape: *(3, 224, 224)*.
- *MaxPooling2D((3, 3))*, stride 2.
- *Residual Block (x2)*: Two convolutional layers with 64 filters, kernel size $3 \times 3$, batch normalization, and skip connections.
- *Residual Block (x2)*: Two convolutional layers with 128 filters. The first block includes downsampling via stride 2.
- *Residual Block (x2)*: Two convolutional layers with 256 filters. Includes downsampling.
- *Residual Block (x2)*: Two convolutional layers with 512 filters. Includes downsampling.
- *Global Average Pooling*: Output size reduced to a 512-dimensional vector.
- *Fully Connected Layer*: Output size 100 (number of classes). Activation: *none (logits)*.

## D.4 HYPERPARAMETERS

For training, we used stochastic gradient descent (SGD) as the optimizer with a learning rate of 0.01 and a momentum of 0.9. Each client trained the model using a local batch size of 64, while the testing batch size was set to 128. These hyperparameters were selected following standard practice in FL literature and remained fixed across all experiments.

## E RESOURCES

To compute each point in Figure 1, Algorithm 2 and Algorithm 3 take approximately 1/2/7 hours for the MNIST/CIFAR-10/CIFAR-100 datasets, respectively. For the parameter reconstruction attack of Algorithm 5, the time required is approximately 20/40/70 hours for MNIST/CIFAR-10/CIFAR-100, respectively. Our proposed method (Algorithm 1) takes around 0.5/1/3 hours for MNIST/CIFAR-10/CIFAR-100, respectively, with nearly all of that time spent on training. We limit training to the

first 10 epochs to reduce unnecessary computational cost, as SIA accuracy tends to be higher in early rounds when client models are more diverse. The experiments were conducted on a server equipped with two NVIDIA RTX 6000 GPUs (24 GB each), two AMD EPYC 7302 16-core processors, 512 GB of RAM, and 10 Gbps Ethernet connectivity.

The other experiments that do not necessitate the use of a GPU (i.e. Table 10 and Figure 2) were computed on a Mac M1 Laptop with a Apple Mac M1 Max chip and 64 GB of memory.

# F    ADDITIONAL EVALUATION

## F.1    OTHER DEFENSES AGAINST SIAS

Hu et al. (2024) discusses that regulaziration-based defences are insufficient against SIAs. In this section we verify this claim by performing experiments on MNIST and CIFAR10 datasets with 10 clients and $\alpha = 0.1$ (the level of heterogeneity). Table 2 shows that all standard regularization approaches fail to address SIAs; rather, they may make the model even slightly more susceptible to them, as they might increase the distributional differences between clients.

Furthermore, we also test Instahide Huang et al. (2020), which was designed for Data Reconstruction Attacks (DRAs). Instahide works by having each client blend their images with some $k$ (a parameter of the mechanism) random, publicly accessible ones. This forms a layer of obfuscation which prohibits the adversary from reconstructing the images. Table 3 shows that this defense is ineffective against SIAs. Instahide focuses on preventing the adversary from reconstructing a particular image, but SIAs consider that the adversary does know that the image exists in the dataset. If the images are blended, with Instahide, then this means that the adversary already knows the blended image. Since this blend is not happening between clients (e.g. shuffling their images) but between each client and a public dataset, the distributional differences are not affected. Note that $k$ in Instahide is small, meaning that only a small portion of the image is altered, to retain model accuracy.

Table 2: Regularization-based defenses against SIAs with 10 clients

| Experiment | Parameter | Dataset | Model Acc. | Model Acc. (vanilla FL) | SIA Acc. | SIA Acc. (vanilla FL) |
|---|---|---|---|---|---|---|
| Random cropping | Crop=24 | MNIST | 92.56 | 97.22 | 50.24 | 45.01 |
| | Crop=20 | MNIST | 83.30 | 97.22 | 53.17 | 45.01 |
| | Crop=16 | MNIST | 62.15 | 97.22 | 56.10 | 45.01 |
| | Crop=28 | CIFAR10 | 58.78 | 61.07 | 52.50 | 51.57 |
| | Crop=24 | CIFAR10 | 53.66 | 61.07 | 53.07 | 51.57 |
| | Crop=20 | CIFAR10 | 44.81 | 61.07 | 59.20 | 51.57 |
| Dropout | p=0.2 | MNIST | 96.45 | 97.22 | 55.70 | 45.01 |
| | p=0.5 | MNIST | 96.28 | 97.22 | 50.92 | 45.01 |
| | p=0.8 | MNIST | 96.42 | 97.22 | 45.17 | 45.01 |
| | p=0.2 | CIFAR10 | 60.76 | 61.07 | 57.77 | 51.57 |
| | p=0.5 | CIFAR10 | 60.38 | 61.07 | 55.20 | 51.57 |
| | p=0.8 | CIFAR10 | 60.52 | 61.07 | 54.54 | 51.57 |
| Weight decay | e=0.5 | MNIST | 97.34 | 97.22 | 50.94 | 45.01 |
| | e=0.4 | MNIST | 97.32 | 97.22 | 52.87 | 45.01 |
| | e=0.3 | MNIST | 97.00 | 97.22 | 55.27 | 45.01 |
| | e=0.5 | CIFAR10 | 61.01 | 61.07 | 56.89 | 51.57 |
| | e=0.4 | CIFAR10 | 61.02 | 61.07 | 57.70 | 51.57 |
| | e=0.3 | CIFAR10 | 60.59 | 61.07 | 61.97 | 51.57 |

Table 3: Instahide against SIAs with $\alpha = 0.1$, 10 clients on the MNIST dataset. SIA accuracy on vanilla FL is $45.01\%$ and vanilla model accuracy is $97.22\%$

| $k$ | Model Accuracy | SIA Accuracy |
|-----|----------------|--------------|
| 1 | 97.48 | 50.3 |
| 2 | 86.65 | 49.9 |
| 3 | 81.52 | 50.9 |
| 4 | 78.65 | 50.45 |
| 5 | 78.37 | 52.71 |

### F.2 PROTECTION FROM SIAS

#### F.2.1 DIFFERENT HYPERPARAMETERS

Figure 6 shows the success rate of SIA for Algorithm 1 and the reconstruction Algorithms 2, 3 and 5 with varying number of clients when $\alpha$ is fixed to 0.1 with 10 local epochs. We observe that the success rate depends on the number of clients which clarifies the reason why SIAs are more of a threat in the cross-silo setting. In all cases, model-level and layer-level shuffling offer insufficient protection, while parameter-level shuffling performs better but the SIA accuracy is still better than random guessing. In contrast our proposed method of Algorithm 1 offers robust protection.

Moreover, we conduct an experiment where we vary the number of local epochs (Figure 7), which does not yield notably different results because of the small $\alpha$ (i.e. datasets are already easily distinguishable).

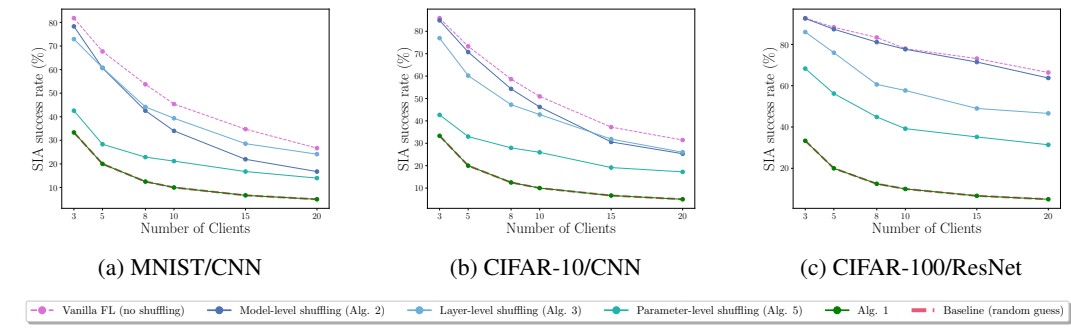

(a) MNIST/CNN      (b) CIFAR-10/CNN      (c) CIFAR-100/ResNet

Figure 6: Success rate of SIA for varying number of clients when $\alpha = 0.1$ with 10 local epochs

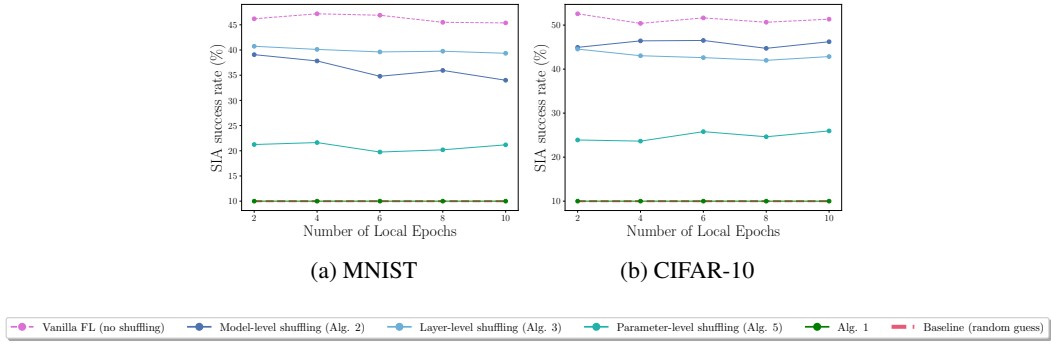

(a) MNIST      (b) CIFAR-10

Figure 7: SIA success rate across different number of local epochs with 10 clients and $\alpha = 0.1$

Finally, we examine how the accuracy of the Reconstruction Attacks of Section 5 is affected when the adversary has only an "approximate" shadow dataset. In this setting, instead of exactly knowing a few data points, the adversary knows a noisy version of them. We run experiments on CIFAR-10 ($\alpha = 0.1$, 10 clients, 10 global epochs, 10 local epochs) using two noise types: geometric noise (Table 4) and color noise (Table 5). Both tables show that standard shuffling techniques fail to guarantee perfect

protection (i.e., equal to random guessing, which is $10\%$ in this case). In contrast, our proposed approach consistently provides robust protection, as our theoretical guarantees are independent of the adversary's side knowledge (Theorem 1).

Table 4: SIA success rate (%) on a shadow dataset with geometrical noise

| Method | Original | Good (30° rotation) | Mid (H-Flip) | Bad (V-Flip) |
|---|---|---|---|---|
| Model-level shuffling (Alg. 2) | 45 | 40.6 | 32 | 26 |
| Layer-level shuffling (Alg. 3) | 42 | 33 | 27.3 | 25 |
| Parameter-level shuffling (Alg. 5) | 27 | 24.1 | 21.7 | 19.3 |
| Proposed Solution (Alg. 1) | 10 | 10 | 10 | 10 |

Table 5: SIA success rate (%) on a shadow dataset with color noise (brightness = contrast = saturation = $x$, hue = $x/10$)

| Method | Original | Good ($x = 0.2$) | Mid ($x = 0.4$) | Bad ($x = 0.6$) |
|---|---|---|---|---|
| Model-level shuffling (Alg. 2) | 45 | 31.9 | 22 | 18 |
| Layer-level shuffling (Alg. 3) | 42 | 39 | 18 | 16 |
| Parameter-level shuffling (Alg. 5) | 27 | 19 | 14 | 12 |
| Proposed Solution (Alg. 1) | 10 | 10 | 10 | 10 |

In the experiments of the main body we assumed that the adversary holds a shadow dataset that is $5\%$ of the one actually used. Table 6 shows that even when this percentage is smaller, the adversary still has an accuracy greater than random guessing (which is $10\%$).

Table 6: SIA success rate for a smaller shadow dataset (CIFAR-10/CNN, $n = 10$, $\alpha = 0.1$)

| Method | 0.5% shadow db | 1% shadow db |
|---|---|---|
| Vanilla FL | 51.57 | 51.57 |
| Model-level shuffling (Alg. 2) | 21.2 | 25.4 |
| Layer-level shuffling (Alg. 3) | 20.24 | 23.5 |
| Parameter-level shuffling (Alg. 5) | 16.2 | 19.1 |
| Proposed Solution (Alg. 1) | 10 | 10 |

### F.2.2 DIFFERENT MODELS

We conduct experiments the Synthetic tabular dataset of Hu et al. (2024), using the same MLP architecture as Hu et al. (2024) for a fair comparison. Table 7 shows results consistent with the previous experiments, supporting the generality of SIAs. We observe that model-level, layer-level and parameter-level shuffling reduce the success rate of SIAs but fail to completely mitigate the attack. However, the proposed Algorithm 1 shows the same SIA accuracy as random guessing.

To evaluate the proposed method across different data modalities, we conducted additional experiments on a time-series dataset (UCI HAR dataset Reyes-Ortiz et al. (2012)) and a text dataset (the 20 Newsgroups dataset Lang (1995)) in addition to synthetic data. For the UCI HAR dataset, we employ a 1D CNN, and for the 20 Newsgroups dataset, we use DistilBERT Sanh et al. (2019). The experimental results are summarized in Table 8. In the Synthetic and HAR datasets, $r = 4$ can achieve the baseline accuracy. For the 20 Newsgroups dataset, the model accuracy is close to the accuracy of Vanilla FL when we set $r = 3$ and almost matches it when $r = 4$.

Our experimental results show that reducing the model parameters with $r = 4$ is sufficient to achieve a balance between model accuracy and protection against SIAs. Further compression beyond this point provides marginal accuracy benefit while incurring greater communication cost. In conclusion, $r = 4$ preserves accuracy close to that of the vanilla model across all evaluated datasets (MNIST, CIFAR10, CIFAR100, Synthetic, HAR, and 20 Newsgroups), while significantly reducing the success rate of SIA.

We would also like to discuss an alternative approach for LLMs. Instead of scaling by $10^r$ (as in Algorithm 1), one could perform integer quantization. Notably, recent work has shown that each parameter of BERT can be represented as an 8-bit integer with negligible accuracy loss Kim et al. (2021). When such 8-bit quantization is combined with our RNS-based encoding, the resulting co-prime moduli introduce only a $1.28\times$ expansion factor over Vanilla FL's 32-bit binary encoding, while still enabling a full depiction of the quantized model parameters. In other words, the combination of integer quantization with Algorithm 1 achieves $1.28\times$ expansion with exactly the same model accuracy of Kim et al. (2021) (i.e. negligibly less than vanilla FL). With this combination, the SIA accuracy in BERT decreases to 10% (i.e. equivalent to random guessing) compared to the 69.8% SIA success rate of Vanilla FL.

Table 7: SIA success rate on the synthetic dataset of Hu et al. (2021) with MLP

| Method | Success Rate (%) |
|---|---|
| Vanilla FL | 46.2 |
| Model-level shuffling (Alg. 2) | 39.3 |
| Layer-level shuffling (Alg. 3 ) | 37.0 |
| Parameter-level shuffling (Alg. 5 ) | 21.1 |
| Proposed Solution (Alg. 1) | 10.0 |

Table 8: Model Accuracy (top-1) of Alg. 1 with precision $r$ compared to vanilla FL (10 clients) on Synthetic, HAR, and 20 Newsgroups datasets.

| Dataset | Model Accuracy | | | | | SIA Accuracy | |
|---|---|---|---|---|---|---|---|
| | Vanilla FL | $r=1$ | $r=2$ | $r=3$ | $r=4$ | Vanilla FL | Alg. 1 |
| Synthetic | 80.48 | 65.19 | 75.96 | 76.07 | 80.30 | 54.30 | 10 |
| HAR (time-series) | 90.38 | 16.83 | 53.58 | 87.72 | 89.28 | 55.60 | 10 |
| 20 Newsgroups (text) | 67.90 | 3.03 | 5.16 | 64.00 | 66.90 | 69.80 | 10 |

### F.3 PROTECTION AGAINST DATA RECONSTRUCTION ATTACKS

In this section we discuss how our proposed method can effectively defend against DRA. One of the earliest works on DRA, Deep Leakage from Gradients (DLG) Zhu et al. (2019b), assumes that each client uses a very small batch size for local training (at most 8 samples). However, in our approach, all client gradients (or model parameters) are mixed before being transmitted to the server (using Algorithm 1). Thus, even if each client employs a batch size of 1, the aggregated gradient reflects $n$ data points, where $n$ is the number of clients.

We conducted experiments against the DLG attack of Zhu et al. (2019b), which show that our approach can substantially reduce its effectiveness. Specifically, while the reconstruction loss of the original attack is 0.0003, it increases to 0.98 when Algorithm 1 is applied (even with as few as five clients). Thus the reconstructed images under Algorithm 1 become severely degraded, with no recognizable features Figure 8. Other DRA techniques are capable of operating with larger batch sizes (e.g., Yin et al. (2021) employs up to 48 samples per client). However, Yin et al. (2021) also shows that the success rate of the attack decreases as the batch size increases. This suggests that our method, by effectively conveying larger batch sizes, can further help mitigate DRA.

### F.4 OTHER AGGREGATION FUNCTIONS

Federated Proximal (FedProx) Li et al. (2020) adds a proximal term as a regularization penalty during local model optimization. The global model is then computed as the average of the local models. In Federated Stochastic Gradient Descent (FedSGD) McMahan & Moore (2017) clients compute gradients over their data just once per round. The central server collects all local models and averages them.

Figure 9 shows that for both cases SIAs have higher success rate than random guessing for all standard shuffling approaches. In contrast, the proposed framework reduces the accuracy of the attacks to

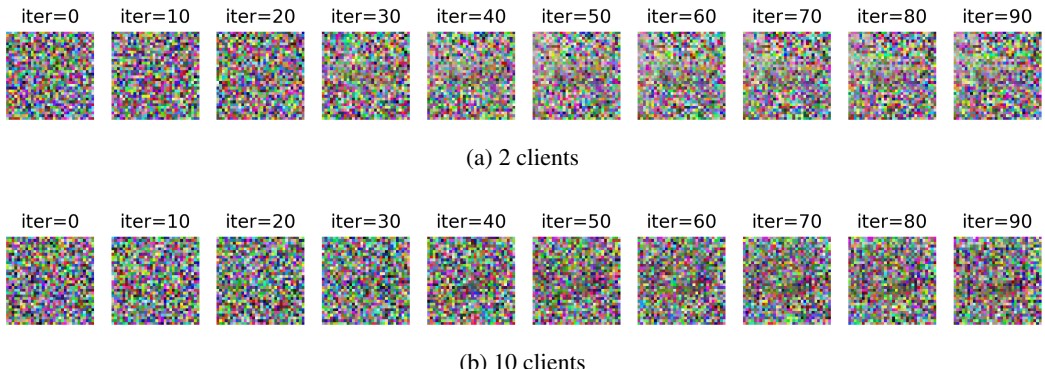

(a) 2 clients

(b) 10 clients

Figure 8: DLG attack on Algorithm 1 (CIFAR-10 with batch size 1)

random guessing. This indicates that our approach can be extended to other sum-based aggregation functions. Figure 10 shows that $r = 3$ suffices to reach an accuracy comparable to vanilla FL and $r = 5$ to match it.

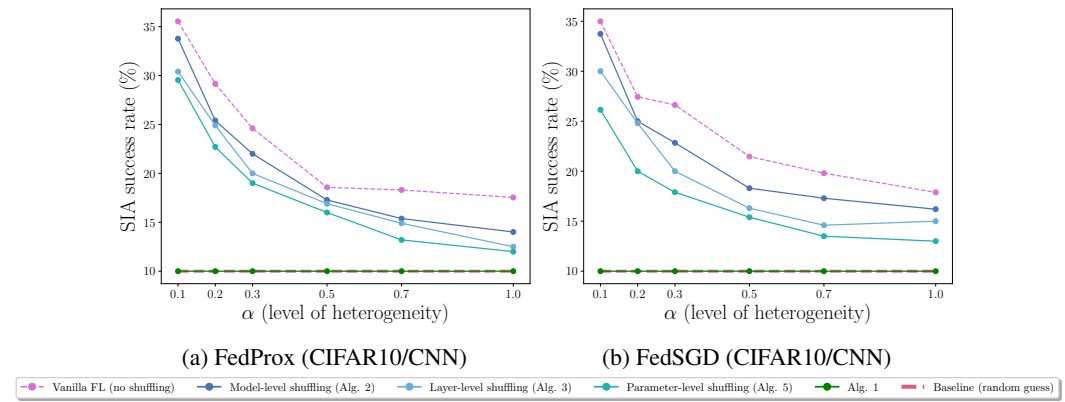

(a) FedProx (CIFAR10/CNN)     (b) FedSGD (CIFAR10/CNN)

Vanilla FL (no shuffling) Model-level shuffling (Alg. 2) Layer-level shuffling (Alg. 3) Parameter-level shuffling (Alg. 5) Alg. 1 Baseline (random guess)

Figure 9: Success rate of SIA for 10 clients on FedProx and FedSGD

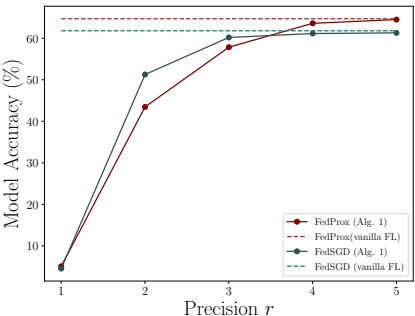

Figure 10: Model Accuracy (top-1) on CIFAR-10/CNN of Alg. 1 with precision $r$ compared to vanilla FL (10 clients).

### F.5 COMMUNICATION COST

Figure 11 shows that RNS can encode even large numbers using a small set of moduli, minimizing the number of shuffling rounds. Moreover, Figure 12 presents a zoomed-in view of Figure 2 for up to 20 clients. For $r = 4$, we observe that Algorithm 1 requires approximately 23 extra bits per parameter

compared to vanilla FL, increasing to around 78 additional bits when $r = 8$. The communication cost of SA is slightly lower only when $r = 8$ and the number of clients is small (i.e. fewer than 5).

Note that Figure 12 shows that Alg. 1 with compression can even surpass standard vanilla FL. However, recall that Alg. 1 transmits only the first $r$ digits of the parameter whereas vanilla FL transmits the whole value. Conversely, if one wants to use the standard binary compression of vanilla FL to transmit only the first $r$ digits, they might be able to use less bits. For example if $r = 3$ than a 12-bit binary encoding is sufficient. In any case, we want to note that the proposed technique does not outperform vanilla FL in terms of communication cost, as it transmits less information.

Table 9 evaluates the proposed technique for more clients and more complex models (necessitating larger $r$). We observe that for both parameters Secure Aggregation scales poorly; in contrast the proposed technique offers reasonable communication cost, especially when combined with compression. Recall that Algorithm 1 assumes a *Semi-Honest* shuffler; if the shuffler is *Fully Trusted* than coupling with RLE is possible.

Table 9: Communication cost for larger parameters (bits per user per parameter)

| Clients | $r$ | SA | Alg. 1 | Alg. 1 + RLE. | Vanilla FL |
|---------|-----|--------|--------|---------------|------------|
| 1000    | 8   | 36926  | 160    | 43            | 32         |
| 10000   | 8   | 399920 | 160    | 42            | 32         |
| 1000    | 12  | 49900  | 281    | 61            | 32         |
| 10000   | 12  | 539892 | 328    | 67            | 32         |
| 1000    | 16  | 63872  | 381    | 73            | 32         |
| 10000   | 16  | 669866 | 440    | 79            | 32         |

Finally, Figure 14 shows the expansion factor, that is the ratio of the encoded model size to the size of the initial vanilla FL model (i.e. using standard 32-bit binary encoding). The plot shows that for 10 clients, an expansion factor of $1.81\times$ is sufficient to achieve nearly the same accuracy as vanilla FL, and $2.4\times$ is needed to fully match it. If RLE compression is used, the expansion factor is $1.03\times$, which is a negligible overhead.

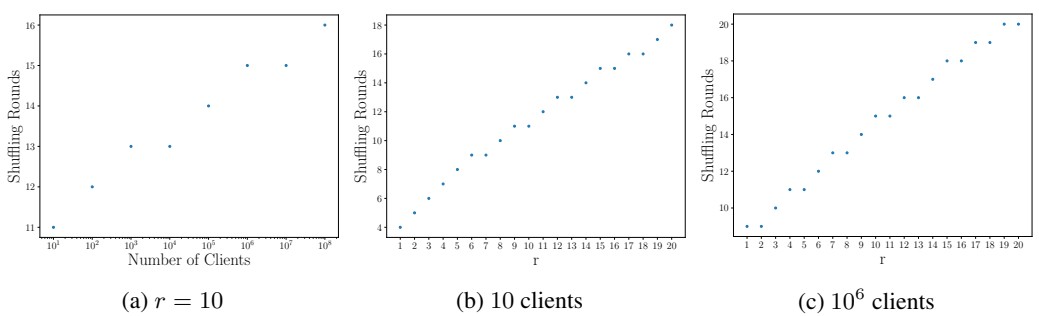

(a) $r = 10$        (b) 10 clients        (c) $10^6$ clients

Figure 11: Number of shuffling rounds in Algorithm 1

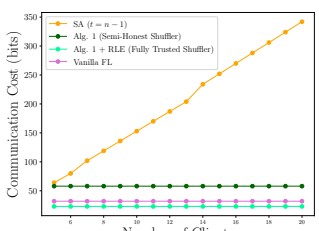

Figure 12: $r = 4$

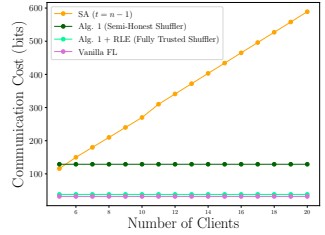

Figure 13: $r = 8$

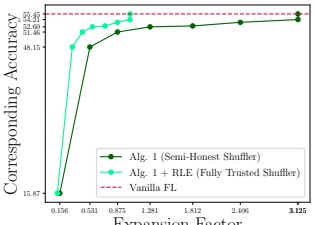

Figure 14: Expansion factor on CIFAR-100/ResNet when $n = 10$

### F.6 COMPUTATION COST

The encoding overhead of Algorithm 1 is negligible for the clients since it relies on fast and primitive operations (modulo and unary encoding). As shown in Table 10, even ResNet can be encoded in just 19.1 seconds, which is negligible compared to the total training time. Similarly, decoding (performed by the central server) is slightly slower but still remains efficient (e.g. slightly less than a minute for ResNet).

Table 10: Computation Time in seconds

| DB/Model | # of Parameters | Encoding | Decoding |
|---|---|---|---|
| MNIST/CNN | 643850 | 1.16 | 3.8 |
| CIFAR-10/CNN | 940362 | 1.6 | 5.6 |
| CIFAR-100/ResNet | 11237432 | 19.1 | 57 |

### F.7 SIA SUCCESS RATE WITH SUBSAMPLING

Figure 15 illustrates the accuracy of SIA when subsampling is applied. In the case of MNIST, the SIA accuracy drops significantly when the number of clients increases from 10 to 50 (from 45% to 10.2%), but rises again when client sampling is applied (44.1% at a 20% sampling rate). Similar trends are observed for CIFAR-10 and CIFAR-100, where SIA accuracy decreases with more clients (from 51% to 16% in CIFAR-10, and from 77% to 26% in CIFAR-100), but increases when only a subset of clients is sampled (up to 50.33% and 72.5%, respectively). This experiment highlights that SIAs can be generalized to settings with more clients, if subsampling is applied.

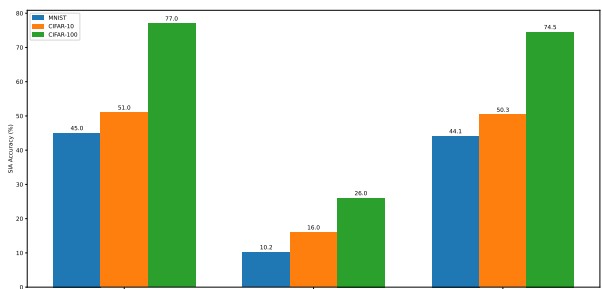

Figure 15: SIA accuracy on MNIST (blue), CIFAR-10 (orange) and CIFAR-100 (green) under client sampling with $\alpha = 0.1$ and 10 local epochs.

F.8    SYNERGY WITH DP

**Privacy of DP-SGD**    A well-known property of DP is that post-processing does not affect the privacy guarantees of a mechanism:

**Lemma 2** (Post-Processing). *Dwork & Roth (2014) If $M$ is $(\varepsilon, \delta)$ - approximate differentially private then for every function $A$, $A \circ M$ is $(\varepsilon, \delta)$ - approximate differentially private.*

Combining Algorithm 1 with DP-SGD, involves encoding the parameters (using RNS) after they have been obfuscated with DP. Hence, encoding is post-processing (i.e. performed after the addition of noise) and thus the privacy analysis of DP-SGD is not affected.

However, Algorithm 1 is based on the assumption that the parameters of the model can be bounded; in our work, we assume w.l.o.g. that every parameter $p$ is in (-1,1). DP-SGD Abadi et al. (2016) bounds (i.e. clips) the gradient but then adds unbounded Gaussian noise. While clipping does help, the noisy output might, theoretically, fall outside of (-1,1).

To avoid this problem we can use a truncated Gaussian; it is known that such mechanism offers the same Rényi-DP (RDP) privacy guarantees as the standard non-truncated Gaussian Fu et al. (2023). DP-SGD measures privacy by the Moments Accountants method, which is essentially a scaled variant of RDP Abadi et al. (2016), thus using a truncated Gaussian will not affect its privacy analysis. Furthermore, the truncated Gaussian mechanism also satisfies the same $\varepsilon$ DP-guarantee as the unbounded version Chen (2024).

In case one wishes to use something other than the standard Gaussian mechanism for DP-SGD, one solution would be to bound the probability that the mechanism outputs an out-of-bounds value by the $\delta$ term of DP (e.g. by using Markov's inequality, which is a common technique Athanasiou et al. (2025)). Another approach, to avoid adding $\delta$, would be for the clients to collectively find the bounds after adding noise by calculating their minimum and maximum values (e.g. with an MPC protocol). In that case the communication cost does not depend on the number of dimensions (i.e. number of parameters) as each client will first compute locally their minimum and maximum over all of their dimensions.

In practice, anyway, even with the standard Gaussian, it is very rare to get a result out of range, since the amount of noise added is usually rather small. The standard deviation of the noise $\sigma$ is $C \cdot \text{NoiseMultiplier}/\text{BatchSize}$, which is typically very small. For example, with a clipping norm $C = 1$, noise multiple r =1.0, batch size = 64, the resulting standard deviation is only $\sigma \approx 0.015$. Furthermore, the (noisy) gradient is also multiplied by the learning rate, thus the DP noise is also downscaled.

To empirically validate this behavior, we conducted experiments on the MNIST dataset under three different privacy budgets ($\varepsilon$) using 10 clients, 10 local epochs, 10 global epochs and $\alpha = 0.1$. Table 11 shows that not a single parameter of any client was outside of the range (-1,1).

Table 11: Range of Parameters for different $\varepsilon$

| $\varepsilon$ | Range of Parameters |
|---|---|
| 1 | $[-0.438, \; 0.592]$ |
| 2 | $[-0.412, \; 0.573]$ |
| 5 | $[-0.400, \; 0.543]$ |

**SIA protection**    Regarding protection (i.e. SIA accuracy), coupling DP-SGD with our mechanism does not affect the defense, since Algorithm 1 and DP are orthogonal to each other, in the sense that adding DP noise does not affect the analysis of Theorem 1. We conducted an evaluation combining Algorithm 1 with DP-SGD on MNIST (10 clients, 10 local epochs, 10 global epochs, $\alpha = 0.1$), and the SIA accuracy was $10\%$ (i.e. equal to random guessing).

**Communication Cost**    We also tested if adding DP affects the selection of the parameter $r$ on the MNIST dataset (10 clients, 10 local epochs, 10 global epochs, $\alpha = 0.1$). Table 12 shows that selecting $r = 3$ achieves comparable accuracy and $r = 5$ matches baseline accuracy (i.e. that of combining Vanilla FL with DP-SGD). In comparison, $r = 2$ and $r = 3$ are necessary for Vanilla FL

without DP. This means that adding DP slightly increases the communication cost, as more precision is necessary to accurately depict the noisy parameters.

Table 12: Model Accuracy with different $r$ on MNIST

| Setting | No DP | $\varepsilon = 1$ | $\varepsilon = 2$ | $\varepsilon = 5$ |
|---|---|---|---|---|
| Baseline | 98.2% | 87% | 88.46% | 90.21% |
| Alg. 1 with $r = 1$ | 12% | 11% | 11.48% | 11.65% |
| Alg. 1 with $r = 2$ | 96.13% | 70% | 71.67% | 75.94% |
| Alg. 1 with $r = 3$ | 98% | 83.51% | 84.72% | 87.56% |
| Alg. 1 with $r = 4$ | 98.07% | 85.95% | 87.32% | 89.83% |
| Alg. 1 with $r = 5$ | 98.12% | 86.44% | 88.07% | 90.12% |

In conclusion, the synergy offers the same privacy guarantees, albeit with a slightly higher communication cost to achieve baseline-level accuracy.

### F.9 EXPANSION TO NON-SUM-BASED FUNCTIONS

Algorithm 1 reveals only the aggregated model, which works well with sum-based aggregation functions (e.g. FedAvg, FedSGD, FedProx). A natural question, however, arises: *what if the central server wishes to compute something other than the sum or the average*? Consider, for example, FedMedian, where the server outputs the median of each parameter. To compute the median, the server needs access to all the parameters, not just their sum. Thus, the threat of SIAs emerges again. Since model obfuscation is not a feasible approach (cf. Appendix F), we must once more resort to our proposed method of Algorithm 1 to provide protection.

Instead of shuffling the gradients of all $n$ clients (as in Algorithm 1), we can group them into small groups (clusters) of $k$ clients and shuffle the gradients within each group before sending them to the server. From the analysis of Algorithm 1 we know that only the aggregated group model is disclosed. For instance, with $n = 10$ and $k = 2$, the server receives 5 shuffled groups. The server can then apply the FedMedian algorithm to these 5 aggregated groups (naturally, when $k = n$, this is the same as FedAvg). An SIA can at most determine which group a value belongs to, but not the specific client. The clients themselves can coordinate the formation of clusters by using a secure channel (e.g. MPC) to decide which peers join each group. The central server remains oblivious to the client composition of every cluster. Therefore, learning the cluster to which a client belongs does not reveal any sensitive information.

We conducted preliminary experiments on MNIST with $n = 10$, $k = 2$, $\alpha = 0.1$, and $r = 2$, selecting the clusters at random. The resulting model accuracy ($97.2\%$) was nearly identical to vanilla FedMedian ($\approx 98\%$). Moreover, under this approach the SIA accuracy remained at random-guessing levels (i.e. $10\%$). Thus, mixing as few as two clients' gradients might be sufficient to prevent SIAs, while still allowing the server to approximate the median using aggregated information from the small client groups.

We aim to further explore such approaches for other-sum-based functions as part of our future work.

