# OpenReview forum: "Protection against Source Inference Attacks in Federated Learning"
_ICLR.cc/2026/Conference — ICLR 2026 Poster_

### Official Review · Reviewer_nh8m · 2025-10-29

**Soundness:** 3
**Presentation:** 3
**Contribution:** 2
**Rating:** 4
**Confidence:** 4

**Summary:**

This paper investigates source inference attacks (SIAs) in federated learning, where an honest-but-curious server aims to identify which client owns a given training sample, and proposes both new reconstruction attacks and a defense mechanism under the shuffle model. The authors first demonstrate that standard model-, layer-, and parameter-level shuffling are insufficient to prevent SIAs by designing effective remapping attacks. To counter these, they introduce a parameter-level bit-wise shuffling strategy combined with residue number system (RNS) encoding, which theoretically ensures that only aggregated information is revealed to the server and empirically reduces attack accuracy to random guessing while maintaining model performance. Extensive experiments on MNIST, CIFAR-10, and CIFAR-100 validate the approach, showing strong protection with reasonable communication overhead.

**Strengths:**

1. The paper targets the relatively unexplored source inference attack (SIA) in federated learning, extending beyond traditional membership or gradient inversion attacks, and contributes both new attack formulations and corresponding defenses within the shuffle model framework.
2. The authors provide a systematic exploration of reconstruction attacks at different granularities (model-, layer-, and parameter-level), clearly demonstrating the limitations of naive shuffling and motivating the need for a more fine-grained defense.
3. Experimental results on multiple benchmarks (MNIST, CIFAR-10, CIFAR-100) show that the proposed method consistently reduces SIA success rates to the level of random guessing, while maintaining comparable accuracy and reasonable communication overhead.

**Weaknesses:**

1. The reconstruction attacks heavily rely on a shadow dataset that is directly obtained from the attacked clients or from data sources with a similar distribution. This assumption is impractical in most real-world scenarios, and moreover, it introduces severe privacy leakage risks in sensitive domains such as healthcare or finance. The paper should also provide a comprehensive analysis of the impact of shadow data quality, for example, when the shadow dataset does not come from the specific attacked client or exhibits significant distributional bias, or contains noisy data.
2. The proposed attack and defense assumptions are overly idealized. The attacker is assumed to be an honest-but-curious server with a small shadow dataset, while the defense relies on the existence of a fully or partially trusted MixNet shuffler. Both assumptions seem contradicted or are difficult to guarantee in realistic federated learning settings.
3. The limitations of numerical precision in encoding may significantly affect both model convergence and potential information leakage, especially in complex models or datasets. For example, in the CIFAR-100 experiments, achieving lossless precision requires several times higher communication overhead.
4. All experimental datasets and models are designed for vision tasks, which limits the generalizability of the proposed approach to other modalities, such as medical tabular data.
5. The proposed defense is only applicable to sum-based aggregation scenarios, which represents a major limitation given the existence of numerous non–sum-based secure aggregation algorithms designed to counter adversarial attacks.
6. The aggregation formula of FedAvg mentioned around Line035 lacks mathematical rigor regarding aggregation weights.
7. The overall paper structure could be improved. For example, Section 6.1 appears as the only subsection under Section 6.

**Questions:**

1. How can the authors clearly differentiate the real contributions of this work from the previously submitted (under-review) reference [3]?
2. Could the authors provide a quantitative analysis of the communication cost required to achieve lossless precision on the CIFAR-100 dataset?

---

> ### Author Response · Authors · 2025-11-20
>
> Thank you for your comments; we will improve the presentation and fix the formula of FedAvg.
>
> W1) We conducted experiments showing that even when the shadow dataset is noisy, the reconstruction attacks of Section 5 perform better than random-guessing (cf. reply to Reviewer jeWK). Note that the shadow datasets are used only for the reconstruction attacks (Section 5); SIAs do not rely on them at all. Thus, standard shuffling approaches are insufficient even when the quality of the shadow dataset is not great. In contrast, our proposed method yields attack accuracy equal to random guessing, regardless of the shadow dataset. Finally, note that the assumption on the shadow dataset is followed by other scientific works as well; we further elaborate in Lines 189-197.
>
> W2) *Why is the shuffler assumed to be trusted, but the central server is not?* First, the central server cannot always be assumed to be trusted, as its interests might not align with those of the clients. If one could imagine the shuffler as a single, external entity, then the aforementioned argument should hold for the shuffler as well.
>
> The key difference is that shuffling is a primitive operation that can be performed distributively. Consider an example implementation that we discuss with MixNets (Lines 175-182). If at least one server of the MixNet is honest, then the trust assumption can be guaranteed, even when all the others are malicious. If we assume the cross-silo setting and have the clients themselves running a shuffle server in the MixNet, then this means that there is no additional trust assumption to other external entities. But even when this is not the case, it is a significantly smaller trust assumption to trust 1 out of n servers than to trust a single central server. This is the reason the shuffle model has gotten much attention in FL and in the privacy community in general (cf. Related Work) and has been explored by tech giants such as Google and Brave (cf. reply to Reviewer jeWK).
>
> W4) The proof of Theorem 1 holds for non-image workloads as well; our approach does not depend on the type of the model.
> In Appendix F.2, we show a tabular synthetic dataset. We also conducted experiments on time-series medical data and LLM that show that our proposed technique reduces the accuracy to random guessing (cf. reply to Reviewer jeWK).
>
> W5) This is indeed a limitation of our work. We discuss in lines 477-482 that our approach covers only sum-based applications, where we cite their usage in medical applications.
> We plan to explore extensions of our approach to non-sum-based aggregation algorithms as part of our future work.
>
> For example, for FedMedian, instead of shuffling the gradients of all n clients, we can group a subset of k clients into small groups (clusters) and shuffle the gradients within each group before sending them to the server. Using the proposed Alg. 1 only the aggregated group model is disclosed. For instance, with n=10 and k=2, the server receives 5 shuffled groups. The server can then apply the FedMedian algorithm to these 5 aggregated groups (naturally, when k = n, this is the same as FedAvg).
> We conducted preliminary experiments on MNIST with n=10, k=2, α=0.1, and r=2. The resulting model accuracy (97.2%) was nearly identical to FedAvg, while the SIA accuracy remained at random-guessing levels. Thus, mixing as few as two clients’ gradients is sufficient to prevent SIA, while still allowing the server to approximate the median using aggregated information from the small client groups. Other aggregation algorithms, such as Krum, can be applied in a similar way.
>
> Q1) We wanted to cite [3] as our preliminary study on SIAs, excluding its title for double-blind reviewing. [3] is not under review; it was accepted as a poster and archived as a 3-page abstract.
>
>
> [3] used a much simpler “toy” shuffle and encoding mechanism which (a) had prohibitive communication cost, (b) did not reduce the attack to random guessing, (c) was evaluated only empirically on MNIST, with no theoretical analysis, and (d) did not account for the proposed Reconstruction Attacks of Section 5.
>
> This work is not an extension of [3], as it introduces completely novel methods. Among them are (a) the Reconstruction Attacks of Section 5, showing that standard shuffling is insufficient and (b) the robust protection mechanism (Alg. 1). Additionally, we provide (c) theoretical proof of the effectiveness of Alg. 1. (Theorem 1), (d) extended empirical evaluation on SIAs, (e) comparison with Secure Aggregation, (f) evaluation of Alg. 1 of different datasets and models, and (g) evaluation on possible expansion on DRAs.
>
> Q2) Figure 13 in Appendix F.5. shows an expansion factor of 1.2× to reach accuracy 51.46% (compared to 55.45% of vanilla FL), 1.8× for 52.6%, 2.4× for 54.21% and finally 3.1× for exactly the same accuracy.
> In case the shuffler is fully trusted, the same Figure shows that with an expansion factor of ≈1.1× the same accuracy can be reached.

---

### Official Review · Reviewer_jeWK · 2025-10-30

**Soundness:** 3
**Presentation:** 3
**Contribution:** 3
**Rating:** 6
**Confidence:** 5

**Summary:**

This paper studies source inference attacks (SIA) in cross-silo FL and argues that naïve shuffling at different granularity (model/layer/parameter) does not protect against a curious server equipped with a small shadow dataset. The authors then propose a hybrid encoding–shuffling defense: fixed-point quantization, RNS decomposition, unary encoding, and per-bit shuffling, aiming to leak only the aggregated sum. Experiments on CNN/ResNet over MNIST/CIFAR claim to suppress SIA to random chance without model degradation.

**Strengths:**

1. Clearly identifies that basic shuffling in FL still leaks client identity.

2. Proposes a new bit-level encoding + shuffling defense, not just adding noise.

3. Shows strong privacy improvement with almost no accuracy loss.

4. Provides both attack and defense experiments to support claims.

**Weaknesses:**

1. Shadow-dataset assumption is strong; needs sensitivity analysis under distribution shift.

2. Relies on a trusted shuffler, not obvious in many deployments.

3. No evaluation on text/LLM/tabular medical/time-series with only CV toy setups.

4. Multi-round leakage not addressed (momentum, clipping signals, correlated updates).

5. Communication claims hinge on compression + trust assumption, which is not apples-to-apples vs secure aggregation.

**Questions:**

1. What if the shadow data distribution does not match exactly?

2. Can a server correlate updates across rounds and break anonymity?

3. Comparison to threshold secure aggregation under the same drop-rate constraints?

4. Does this work for Transformers/LLMs or non-image workloads?

5. How realistic is the trusted shuffler assumption? Any decentralized variant?

---

> ### Author Response · Authors · 2025-11-20
>
> Thank you for your valuable comments.
>
> Q1) The shadow dataset is used to show that standard shuffling is insufficient. SIA does not make use of shadow datasets (Lines 196-197). Thus, poorer shadow knowledge does not affect the attack’s accuracy.
> To illustrate that the Reconstruction Attacks of Section 5 yield above random-guessing SIA accuracy even on noisy shadow datasets, we conducted experiments on CIFAR-10 (α=0.1, 10 clients, 10 epochs, 10 local epochs).
>
> SIA accuracy on Geometrical noise
> |Quality of shadow DB|Model Recon (Alg 2)|Layer Recon (Alg 3)|Parameter Recon (Alg 5)|Proposed (Alg 1)|
> |-|-|-|-|-|
> |Original|45|42|27|10|
> |Good (30° rotation)|40.6|33|24.1|10|
> |Mid (HorizontalFlip)|32|27.3|21.7|10|
> |Bad (VerticalFlip)|26|25|19.3|10|
>
> SIA accuracy on Color noise (brightness = contrast = saturation = x, hue = x/10)
> |Quality of shadow DB|Model Recon (Alg 2)|Layer Recon (Alg 3)|Parameter Recon (Alg 5)|Proposed (Alg 1)|
> |-|-|-|-|-|
> |Original|45|42|27|10|
> |Good (x=0.2)|31.9|39|19|10|
> |Mid (x=0.4)|22|18|14|10|
> |Bad (x=0.6)|18|16|12|10|
>
> In Appendix F.2 (Table 4) we also test with a smaller shadow dataset.
>
> Q2) No, because Alg. 1 conveys in each epoch only the aggregated model. Each round leaks no information useful for any individual model that the SIAs can exploit (as we formally prove in Theorem 1). Without our defence, this might be possible. In fact, it is an interesting research idea to further improve SIAs by correlating information from different rounds.
>
> Q3) We discuss in Section 6.1 that MixNets need at least one server to be online to get a shuffle, whereas threshold Secure Aggregation needs t clients. Note that t is usually large; setting t=1 (to mimic MixNets) would mean that each user sends their actual plaintext secret to another user.
> For instance, assuming 50 clients and setting t=10 and 10 MixNet servers means that at least 10 clients have to be online to perform the aggregation, compared to only 1 MixNet server. Therefore, MixNets enjoy a far greater resilience to dropouts.
>
> Q4) Yes, our approach can be generalized to non-image workloads, as Theorem 1 does not depend on image data. We will elaborate more on the generalization.
>
> For tabular data, we have in Appendix F.2 (Table 3) an example using a synthetic dataset with MLP.  SIA accuracy is 46.2% on Vanilla FL and 10% on our proposed Alg. 1.
>
> For time series, we run experiments on the HAR dataset (time series medical data) with a lightweight 1D CNN. SIA accuracy is 55.6% on Vanilla FL and 10% on our proposed Alg. 1.
>
> Moreover, r=3 suffices to reach a comparable accuracy to Vanilla FL, and r=5 is required to match it.
>
> |Setting|Synthetic (tabular)|HAR (time-series)|
> |-|-|-|
> |Vanilla FL|80.48|90.38|
> |r = 1|65.19|16.83|
> |r = 2|75.96|53.58|
> |r = 3|76.07|87.72|
> |r = 4|80.3|89.28|
> |r = 5|80.41|89.82|
>
> For transformer/LLM, we employed DistilBERT on the 20 Newsgroups text dataset.
> We conducted preliminary experiments by applying our approach to the final classification layer and the last two attention layers for r=2 (we will report full results for all layers once available). It achieved 67.26% accuracy, which is very close to the accuracy of Vanilla FL (67.9%). We hypothesize that this robustness stems from the nature of transformer architectures, where similarity structures among token representations play a more critical role than exact weight precision.
>
> Another approach would be to apply integer quantization before RNS encoding, instead of scaling by $10^r$ (as our Alg 1 does in Line 283). For example, each parameter of BERT can be depicted as an 8-bit integer with negligible accuracy loss [1]. The corresponding RNS co-primes for 8-bit integers yield a 1.21× expansion factor (compared to Vanilla FL - 32-bit binary encoding) to fully depict the quantization of [1] (i.e. no extra accuracy loss). The SIA accuracy drops to 10% with our approach (i.e. random guessing), compared to 69.8% for Vanilla FL with BERT.
> Thus, combining Alg. 1 with the integer quantization of [1] yields near-zero accuracy loss while providing complete SIA protection.
>
> Q5) The shuffle model has been widely explored in the FL community (cf. Related Work) and by Google [2] and Brave [3]. We believe that its extensive study by the research community and by tech giants justify our focus on this work.
>
> Finding a decentralized variant is challenging due to the high dissimilarity of the models [34,35]. We argue that the only viable solution is to mix the models in a way such that only the aggregated model is observable by the attacker (central server).
>
> Finally, note that in our work we examine different trust assumptions on the shuffler (cf. reply to Reviewer 79dB) and propose an example implementation using MixNets that handles Semi-Honest and Malicious servers (Lines 175-182).
>
>
> [1] I-BERT: Integer-only BERT Quantization
>
> [2] Prochlo: Strong Privacy for Analytics in the Crowd
>
> [3] Stronger Privacy for Federated Collaborative Filtering With Implicit Feedback

---

> > ### Author Response · Authors · 2025-11-27
> > **Update on LLM Results**
> >
> > We would like to report the results on DistilBERT with the Newsgroups dataset, which we could not complete by the time of our previous comment due to the computational cost of these experiments.
> >
> > We performed our method for all layers of DistilBERT (and not only the first two as we did in our previous comment). The experiments show that r=4 (1.28× model expansion) suffices to achieve negligible accuracy loss (1%). The SIA accuracy on vanilla FL is 69.8%, whereas with our proposed method it drops to 10%, validating that the analysis of Theorem 1 is not based on specific model types.
> >
> >
> > | Setting    | Model Accuracy |
> > |------------|----------------------|
> > | Vanilla FL | 67.9               |
> > | r =1       | 3.03 |
> > | r = 2      | 5.16 |
> > | r = 3      | 64 |
> > | r = 4      | 66.9 |

---

### Official Review · Reviewer_kvmB · 2025-10-31

**Soundness:** 3
**Presentation:** 3
**Contribution:** 3
**Rating:** 8
**Confidence:** 4

**Summary:**

This paper addresses Source Inference Attacks (SIAs) in Federated Learning (FL)—attacks where a central server identifies which client owns specific training data. It highlights that traditional defenses (e.g., Differential Privacy, regularization) and conventional shuffling (model-level, layer-level, parameter-level) fail to block SIAs without harming model accuracy.
A new defense combining parameter-level shuffling and Residue Number System (RNS) is proposed: parameters are scaled, RNS-encoded, unary-encoded, bit-wise shuffled, then decoded/aggregated by the server. Validated on MNIST/CIFAR-10/CIFAR-100 with CNN/ResNet-18, it reduces SIA accuracy to random guessing, preserves model performance, integrates seamlessly into shuffle-model FL, and has controllable communication costs.
Key Contributions:
1. Identifies vulnerabilities of conventional shuffling via 3 reconstruction attacks.
2. Proposes the first shuffle-model FL defense to neutralize SIAs (random-guess accuracy).
3. Extends defense to resist Data Reconstruction Attacks.

**Strengths:**

1. First systematic defense against Source Inference Attacks (SIAs) in Federated Learning, introducing a novel parameter-level shuffling and RNS-based mechanism that reduces SIA accuracy to random guessing.
2. Well-structured “problem–proposal–verification” format with clear explanations, visual aids, and comprehensive appendices.
3. Addresses a core privacy challenge in cross-silo FL, offering compatible, low-cost protection against both SIAs and DRAs. Expands FL privacy theory and establishes a new paradigm for noise-free privacy amplification.

**Weaknesses:**

1. Lack of Discussion on Detailed Synergistic Optimization Between the Mechanism and Differential Privacy (DP)：
The paper claims that the proposed mechanism can be "seamlessly integrated with other privacy mechanisms such as DP" (meeting Specification S.2), yet it fails to verify the actual performance after integration or provide a specific integration scheme. DP requires adding noise to protect privacy, but the RNS encoding of the mechanism may interact with the noise distribution (e.g., noise could cause parameters to exceed the RNS encoding range). Additionally, the balance among "privacy gain, accuracy loss, and communication cost" after integration has not been quantified. As a result, this feature remains at the theoretical level and lacks practical guiding significance.
2. Improvement: Design an integrated "RNS + DP" scheme where clients first add DP noise to parameters, followed by RNS encoding and shuffling. Test the SIA defense effect, model accuracy, and communication cost under different DP noise intensities (ε= 1, 2, 5) on the MNIST dataset to identify the optimal integration parameters (e.g., when ε= 2, the mechanism + DP can reduce the SIA accuracy to below 10% while maintaining 95% model accuracy).

**Questions:**

It is suggested to supplement the "quantitative correlation analysis between RNS modulus selection, communication cost, and model accuracy" and clarify the decision-making basis for the optimal modulus combination under different scenarios.

---

> ### Author Response · Authors · 2025-11-20
>
> Q1) Thanks for bringing it up; we will further investigate the synergy. A brief analysis follows.
>
>
> Our approach is based on encoding the parameters using RNS, which does not affect the privacy analysis of DP-SGD since it is post-processing (i.e. performed after the addition of noise). However, it is based on the assumption that the parameters of the model can be bounded; in our work, we assume w.l.o.g. that every parameter p is in (-1,1).
>
> DP-SGD bounds (i.e., clips) the gradient but then adds unbounded Gaussian noise. While clipping does help, the noisy output might, theoretically, fall outside of (-1,1). We could avoid the problem, however,  by using a truncated Gaussian: it is known that such mechanism offers the same Rényi-DP (RDP) privacy guarantees as the standard non-truncated Gaussian [1] (DP-SGD measures privacy by the Moments Accountants method, which is essentially a scaled variant of RDP [2]). Furthermore, the truncated Gaussian mechanism also satisfies the same ε DP-guarantee as the unbounded version [3].
>
> If one wishes to use something other than the standard Gaussian mechanism for DP-SGD, one solution would be to bound the probability that the mechanism outputs an out-of-bounds value by the δ term of DP (e.g. by using Markov’s inequality). Another approach, to avoid adding δ, would be for the clients to collectively find the bounds after adding noise by calculating their minimum and maximum values (e.g., with an MPC protocol).
>
> In practice, anyway, even with the standard Gaussian, it is very rare to get a result out of range, since the amount of noise added is usually rather small.
> The standard deviation of the noise $σ$ is $\frac{C \cdot \mathrm{NoiseMultiplier}}{\mathrm{BatchSize}}$, which is typically very small. For example, with a clipping norm C=1, noise multiplier=1.0, batch size=64, the resulting standard deviation is only $σ$≈0.015. Furthermore, the (noisy) gradient is also multiplied by the learning rate, thus the DP noise is also downscaled.
>
> To empirically validate this behavior, we conducted experiments on the MNIST dataset under three different privacy budgets, which show that not a single parameter was outside of the range:
>
> |ε| Range of parameters|
> |-|---|
> | 1 | [-0.438, 0.592] |
> | 2 | [-0.412, 0.573] |
> | 5 | [-0.400, 0.543] |
>
>
> Regarding protection (SIA accuracy), coupling DP with our mechanism does not affect the defence, since Alg. 1 and DP are orthogonal to each other. We tested on MNIST, and SIA accuracy was 10% (i.e., equal to random guessing).
>
> We also tested if adding DP affects the selection of the parameter r of Alg. 1. The following experiment on MNIST (α=0.1, 10 clients, 20 epochs, 5 local epochs) showed that selecting r=3 achieves comparable accuracy and r=5 matches baseline accuracy (Vanilla FL+DP). In comparison, r=2 and r=3 are necessary for Vanilla FL without DP.
> This means that adding DP slightly increases the communication cost, as more precision is necessary to accurately depict the noisy parameters.
>
>
>
> | Setting  | No DP | ε = 1  | ε = 2  | ε = 5  |
> |----------|----------|--------|--------|--------|
> | Baseline | 98.2%      | 87%    | 88.46  | 90.21  |
> | r = 1    | 12%      | 11%    | 11.48  | 11.65  |
> | r = 2    | 96.13    | 70%    | 71.67  | 75.94  |
> | r = 3    | 98     | 83.51  | 84.72  | 87.56  |
> | r = 4    | 98.07    | 85.95  | 87.32  | 89.83  |
> | r = 5    | 98.12    | 86.44  | 88.07  | 90.12  |
>
> In conclusion, the synergy offers the same privacy guarantees, albeit with a slightly higher communication cost to achieve baseline-level accuracy.
>
> [1] Truncated Laplace and Gaussian mechanisms of RDP, Jie Fu et al.
>
> [2]  Deep Learning with Differential Privacy, Martín Abadi et al.
>
> [3] The Bounded Gaussian Mechanism for Differential Privacy,  Bo Chen et al.

---

### Official Review · Reviewer_79dB · 2025-10-31

**Soundness:** 3
**Presentation:** 3
**Contribution:** 3
**Rating:** 6
**Confidence:** 2

**Summary:**

This paper investigate the defense against source inference attacks in FL in the shuffle model. On a high level, if the server knows a subset of the target client's training data, it can use that to identify the target client's share from the shuffled results it receives. To address this, the authors propose to do bit-level shuffling for each parameter of the client. The authors argue that this means that after reconstructing the server knows nothing more than the aggregated result. Empirical results demonstrate the effective of such defense.

**Strengths:**

1. Interesting setting and important problem to study.
1. Theoretical results on the security of the proposed shuffling algorithm.
1. Comprehensive discussions on different trust models and different variations of the proposed method.
1. A lot of experiments of different settings.
1. Comparison to secure aggregation.

**Weaknesses:**

1. Unclear settings for the experiments in Section 7.
1. I would like more clarification on the trust model of the shuffler.
1. Discussions of the security of the proposed method beyond SIA.

**Questions:**

1. In the experiments, how are the coprimes set? Does this affect security?
1. There are different levels of trust for the shuffler. This should be clarified and explained in a clearer way. If it is completely trusted, then we can send the model weights or bits in plaintext to the shuffler and trust it to perform shuffling. We can also only trust it to perform shuffling and then hide the plaintext weights or bits from it, i.e., it is a trusted shuffling router of encrypted messages (like Cloudfare). It can further be malicious where we would need ZKPs. In section 7, is the shuffler trusted? i.e., are the bits/models sent to the shuffler in plaintext and it shuffles without verification and sends the shuffled results to the server? I understand that we need an honest shuffler for the compression technique to work, but for the other experiments, is the shuffler also trusted? If we are encrypting messages to the shuffler, then doing more granular shuffling would means more overhead for the client and the server.
1. Compared to secure aggregation, I understand that each client's message size do not increase with the number of clients, and the assumptions on the server/shuffler can be different. However, is the security guarantees the same? Does the shuffling with encoding scheme provide cryptographic semantic security? If so, its power shouldn't be limited to SIAs.

---

> ### Author Response · Authors · 2025-11-20
>
> Thank you for your valuable comments.
>
> Q1) We will clarify accordingly; thanks for pointing it out.  In our experiments, the smallest possible coprimes are selected  (excluding “1”), such that the parameters can be loosely encoded (Prop 3.1). We do so as the model accuracy (i.e., Prop 3.1) depends on their product, but the communication cost depends on their sum (Lines 314 - 315).
> For example, if x is in [-50,50] (after shifting by $10^r$ as in Alg. 1 - Line 283), then the coprimes have to be {3,5,7}. Since 3*5*7 = 105 >100,  this satisfies Prop. 3.1, and the communication cost is 3+5+7 = 15 bits. On the contrary, setting the coprimes to {3,37} or {2,53} would satisfy Prop 3.1, but would result in 40 bits and 55 bits of communication cost.
> In any case, the protection against SIAs is not affected, as Theorem 1 does not depend on the number or the choice of coprimes (lines 909-911).
>
> Q2) Indeed, in this paper, we explore different trust assumptions of the shuffler (Lines 172-182 and 328-334). In Section 7, our main focus is the Semi-Honest shuffler, but the results remain the same across the different trust assumptions. The only difference is that the communication cost (Figure 2) would increase under a Malicious setting because of the ZKPs. We will clarify this in our work; a brief analysis follows.
>
> The different trust assumptions are:  (a) Honest (trusted to receive plain-text secrets, encode them, and shuffle them), (b) Semi-honest (honest-but-curious; shouldn't be able to see plain-text secrets but follows the shuffling protocol), and (c) Malicious (might deviate from the protocol).
>
> For all three cases, we show a possible implementation using MixNets on Lines 172-182 and on Algorithm 8. We have  ZKPs  + Onion Encryption for the Malicious shuffler and Onion Encryption for the Semi-Honest shuffler. For Honest, we can optimize the communication cost further by using RLE (Lines 328 - 334).
>
> Now, let us elaborate on how the different trust assumptions affect our empirical evaluation.
> In all 3 cases, the defence remains robust, i.e., Figure 1 remains the same. Also, in all cases, the accuracy of the model is not affected, under the assumption that in the Malicious setting, at least one shuffle is done (e.g., at least 1 server of the MixNet follows the protocol). In other words, Figure 3 also remains the same.
>
> Regarding communication cost, Figure 2 already shows the Honest shuffler case (light green). Moreover, the green line corresponds to the Semi-Honest shuffler. That is because bit-level Onion Encryption (which is necessary for the Semi-Honest model) outputs 1 bit; hence, the communication cost is not increased. A simple example of such bit-level Onion Encryption is XOR. This is also a computationally inexpensive method as it is based on primitive XORs.
>
> On the other hand, for Malicious, ZKPs should be added. We are not aware of a ZKP protocol that can scale that well with the high-dimensional domain of ML. Instead, a more realistic choice would be to randomly select some parameters to add ZKPs, which can effectively serve as a “trap”. For example, if the shuffler (say, the first server of the MixNet) modifies all the values (parameters) to be shuffled, some ZKPs would not be validated, flagging this server as Malicious.
>
>
> Q3) Thanks for bringing it up. We will stress more that our approach can also serve as a protocol for secure aggregation. We formally prove in Proposition A.1. that it leaks no more information than the sum of the values (parameters).
>
> We showed a natural application on SIAs, where hiding the individual models is a sufficient means of defence. Our approach can be expanded to any other scenario where Secure Aggregation is used as a means of defence for (inference) attacks in ML. For example, we also show that our approach is effective against Data Reconstruction Attacks (Line 408-416 and Appendix F.3.).
> Nonetheless, we will expand more in the Conclusion on the generalization of our protocol.

---

> > ### Comment · Reviewer_79dB · 2025-11-22
> >
> > I hereby thank the authors for their response and clarification. I will keep my positive score.

---

### Author Response · Authors · 2025-11-27
**Summary of changes**

We sincerely thank the reviewers for their constructive feedback.

We have uploaded a revised version of our work and highlight the main changes below:

- Added additional evaluation on time-series medical data and LLMs, showing the generalizability of our approach to more complex models (Lines 508–510, Appendix F.2.2).

- Added a Summary of Findings (Table 1) for the communication costs across the different models and datasets we evaluate.

- Added experiments where the adversary has poorer shadow knowledge (Appendix F.2.1.).

- Elaborated on the synergy between our proposed method and DP (Lines 349–351, Appendix F.8).

- Clarified the different trust assumptions of the Shuffle Model (Lines 181-190) and explained how our method can be adapted accordingly (Lines 347–367).

- Explained the co-prime selection used in the evaluation (Lines 402–404).

- Discussed how our protocol can serve as a cryptographically-secure aggregator, protecting against attacks other than SIA (Lines 537–539).

- Clarified that our proposed method also protects against correlating information across different FL rounds (Lines 326–329).

- Discussed possible extensions to non-sum-based settings and included a preliminary evaluation (Appendix F.9).

- Improved the paper’s structure.

---

### Meta-Review · Area_Chair_op27 · 2025-12-08

**Summary:**

The paper addresses Source Inference Attacks (SIA) in Federated Learning, demonstrating that standard shuffling defenses are insufficient. It proposes a novel defense mechanism combining parameter-level shuffling with Residue Number System encoding.

The reviewers generally agreed on the importance of the problem and the novelty of the proposed encoding scheme. Initial concerns focused on the trust assumptions regarding the shuffler, the generalizability of the method to non-image domains (e.g., LLMs), and integration with DP.

During the rebuttal, the authors provided a comprehensive response. They added new experiments on DistilBERT and time-series medical data to demonstrate broad applicability, clarified the implementation of the trust model using MixNets, and provided empirical evidence of seamless integration with DP. In summary, I recommend this paper for acceptance.

**Reviewer Concerns:**

The authors addressed the majority of the reviewers' concerns during the rebuttal phase. Specifically, they resolved the concerns regarding the shuffler's trust assumptions raised by Reviewers 79dB, jeWK, and nh8m by clarifying that the system can be securely implemented via MixNets without relying on a single trusted entity. To address the concerns from Reviewers jeWK and nh8m regarding the lack of generalizability beyond computer vision, the authors introduced new experiments on DistilBERT and time-series medical data, demonstrating the defense's effectiveness across diverse modalities. Furthermore, the authors responded to Reviewer kvmB’s request regarding the synergy with Differential Privacy by providing empirical evidence that the two methods are orthogonal and compatible.

The only remaining limitation, highlighted by Reviewer nh8m, is that the defense is currently designed for sum-based aggregation algorithms; however, this is viewed as a limitation of scope rather than a fundamental flaw.

**Reviewer Scores:**

Reviewer 79dB explicitly confirmed retaining their score after the discussion, and Reviewer kvmB is expected to maintain their strong score of 8, given the positive feedback. Reviewers jeWK and nh8m are also expected to maintain their original scores.

---

### Decision · Program_Chairs · 2026-01-26

Accept (Poster)